



# Impact of structure on the estimation of atmospherically relevant physicochemical parameters

Gabriel Isaacman-VanWertz[1] , Bernard Aumont[2]

[1] Department of Civil and Environmental Engineering, Virginia Tech, Blacksburg, VA, 24060

[2] LISA, UMR CNRS 7583, Université Paris-Est-Créteil, Université de Paris, Institut Pierre Simon Laplace, Créteil, France

*Correspondence to*: Gabriel Isaacman-VanWertz (ivw@vt.edu)

**Abstract**. Many methods are currently available to estimate physicochemical properties of atmospherically relevant compounds. Though a substantial body of literature has focused on the development and intercomparison of methods based on molecular structure, there has been an increasing focus on methods based only on molecular formula. However, prior work

has not quantified the extent to which isomers of the same formula may differ in their properties, or, relatedly, the extent to which lacking or ignoring molecular structure degrades estimates of parameters. Such an evaluation is complicated by the fact that structure-based methods bear significant uncertainty and are typically not well constrained for atmospherically-relevant molecules. Using species produced in the modeled atmospheric oxidation of three representative atmospheric hydrocarbons, we demonstrate here that differences between isomers are greater than differences between methods. Specifically, isomers

tend to differ in their vapor pressures and Henry's Law Constants by a half to a full order of magnitude greater than differences between estimation methods, and differ in their $k_{OH}$ by a factor of two. Formula-based estimation of these parameters is shown to be possible with little bias and an approximately normally distributed error. Specifically vapor pressure can be estimated using a combination of two existing methods, Henry's Law Constants can be estimated based on vapor pressure, and $k_{OH}$ can be approximated as a constant for all formulas containing a given set of elements. Formula-based estimation is therefore

reasonable when applied to a mixture of isomers, but creates uncertainty commensurate with the lack of structural information.

## 1 Introduction

The fate of an organic compound in the atmosphere is dictated by a number of physicochemical properties. Its volatility controls whether it partitions to suspended particulate mass or remains in the gas phase, its reactivity controls its lifetime against degradation by ever-present oxidants, and its solubility may control its uptake to particles or its deposition to surfaces

(leaves, soil, etc.). The parameters that describe these properties (e.g. vapor pressure) are consequently a critical term in models describing the physical and chemical transformations of atmospheric constituents. In some cases, an exact estimation of these parameters may not be important; for instance, a compound will almost certainly condense when given the opportunity whether its vapor pressure is extremely low, or merely very low. However, many compounds exist in transition regimes in environments typical of atmospheric conditions in which they can partition between phase and fates, such as: semivolatile compounds that



partition between the gas and particle phase (Donahue et al., 2006); compounds with moderate reactivity that may last hours or days depending on oxidant concentrations (Price et al., 2019); or compounds with sufficient solubility to partition to particles with an aqueous phase but not dry particles (Wania et al., 2015). For these atmospheric components (which likely account for at least tens of percent of atmospheric organic carbon (Hunter et al., 2017)), an accurate estimate of its physicochemical parameters is critical.

Unfortunately, physicochemical parameters for atmospherically-relevant compounds are poorly constrained by experimental data. Vapor pressures and Henry's Law Constants (HLC) are known primarily for higher volatility compounds, typically with few (1-3) functional groups (Compernolle et al., 2011; Raventos-Duran et al., 2010). Little observational data exists for, e.g., compounds with vapor pressures sufficiently low to partition under typical atmospheric conditions. In contrast, the atmosphere contains thousands or tens of thousands of compounds across ~15 orders of magnitude in vapor pressure (Jimenez et al., 2009),

wide ranges of oxygenation, volatility and solubility (e.g., Donahue et al., 2011; Hodzic et al., 2014; Lannuque et al., 2018), and several orders of magnitude in reactivities (Lee et al., 2006), with many multifunctional components (e.g., Aumont et al., 2005; Saunders et al., 2003). Observational databases are consequently of little direct use. In order to estimate these parameters beyond the range of observational constraints, several methods have been developed that relate physicochemical parameters to structure through structure-activity relationships (SARs). These typically take the form of group contribution, in which a

molecular structure is parsed into component groups (carbonyls, esters, carbon-carbon double bonds, etc.) with each group assigned an empirically determined impact on a parameter of interest. Various methods exist to estimate volatility (e.g., Barley and McFiggans, 2010; Camredon and Aumont, 2006; Compernolle et al., 2011), HLC (e.g., Meylan and Howard, 1991; Raventos-Duran et al., 2010) and gas-phase reaction rates (e.g., Vereecken et al., 2018). Previous work focusing on prediction of vapor pressures, HLC and gas-phase reaction rates acknowledge that SAR's estimates for atmospheric species tend to

diverge with increasing number of organic functional groups on the carbon backbone, i.e. for lower volatility and higher water solubility (Valorso et al., 2011). Furthermore, the substituent groups within a complex molecule may not obviously "map" to the groups used to define an SAR, so extrapolation or interpretation may be necessary when applying an SAR to a mixture such as the atmosphere.

Earlier work on vapor pressure implemented a two-step estimation method, in which boiling point is estimated using an SAR,

and vapor pressure is estimated from this boiling point using a separate SAR. Widely used boiling point estimation methods include Stein and Brown (1994), Nannoolal et al. (2004), and Joback and Reid (Joback, 1984; Reid et al., 1987), while widely used vapor pressure estimation methods include Nannoolal et al. (2008) and Myrdal and Yalkowsky (1997). Comparison by Barley and McFiggans (2010) of these eight possible combinations (as well as a few less-widely-used methods) suggest that estimation of boiling point using the Nannoolal et al. (2004) method yields the best agreement with experimental data, in

particular when using the Nannoolal et al. (2008) vapor pressure estimation method. More recently, vapor pressure estimation methods have been developed that use SARs to directly estimate vapor pressure, specifically SIMPOL (Pankow and Asher,



2008) and EVAPORATION (Compernolle et al., 2011). These two methods have been previously shown to agree well with those estimated by the Nannoolal et al. method (Compernolle et al., 2011). Prior work therefore suggests that at least three methods (SIMPOL, EVAPORATION, and Nannoolal) comparably estimate vapor pressures and are in reasonable agreement

with experimental data. However, theses experimental data are mostly limited to vapor pressures greater than $10^{-8}$ atm (saturation concentration, $c* > \sim 10^{1.5}$ μg/m3), which is at the lower limit of vapor pressures expected to partition to the particle phase under typical atmospheric conditions (Donahue et al., 2006). None of these methods was found to be accurate to better than approximately half an order of magnitude for their best constrained regions and methods tend to diverge at lower vapor pressures (Barley and McFiggans, 2010; Compernolle et al., 2011; Valorso et al., 2011). Even relatively accurate estimates

can introduce large errors in transition regimes. An error of half an order of magnitude in vapor pressure for a compound with an estimated saturation concentration near ambient particulate matter concentrations may "move" a compound from mostly in the gas phase to mostly in the particle phase (Compernolle et al., 2011).

For most volatile organic compounds (VOCs), the atmospheric oxidation is mainly driven by the reaction with OH radical. Various methods based on SARs are available in the literature to estimate VOC+OH gas-phase rate constants, $k_{OH}$ (Vereecken

et al., 2018). A very commonly used SAR was developed by Kwok and Atkinson (Kwok and Atkinson, 1995), for which a few revised and extended versions are now available (e.g., Jenkin et al., 2018a, 2018b).

A few methods are available for estimation of HLC, which parameterizes the partitioning of gases into a liquid (typically dilute aqueous) phase. For atmospheric chemistry applications, most commonly used SARs are HWINb (US Environment Protection Agency, 2019) and the more recently developed GROMHE (Raventos-Duran et al., 2010), the latter of which has been shown

to be somewhat more accurate. There is consequently less alternative around selection of a method to estimate these parameters, but there can nevertheless be large errors in their estimation (e.g., orders of magnitude in HLC estimates).

In addition to these methods for estimation of physicochemical parameters based on molecular structure, there has been a recent focus on developing approaches that rely only on molecular formula. This is driven in large part by the rapid increase in the use of direct mass spectrometry, in particular direct chemical ionization mass spectrometry (CIMS), which sample at

atmospheric pressure and can therefore detect nearly all gas- and particle-phase atmospheric constituents with minimal pre-treatment (Aljawhary et al., 2013; Huey et al., 1995; Hunter et al., 2017; Isaacman-vanwertz et al., 2018). By allowing direct measurement of chemically and/or thermally labile atmospheric constituents, these instruments have profoundly increased understanding of atmospheric chemistry (e.g., Ehn et al., 2014; Lee et al., 2016; Nguyen et al., 2015). However, direct mass spectrometry generally lacks any mechanism for the resolution of isomers, yielding data only on the molecular formula of

detected analytes, with little structural information. Some approaches to CIMS are limited to specific compound classes (e.g., acids), thus providing some information, but provide no resolution of isomers within these classes (Thompson et al., 2016). In order to situate measurements by CIMS and other direct mass spectrometers in a chemical space useful for modeling or understanding the atmosphere (e.g., Isaacman-VanWertz et al., 2017; Mohr et al., 2019), methods have been developed and



applied for estimating physicochemical parameters from formulas alone. These methods are primarily limited to estimation of
vapor pressure (Daumit et al., 2013; Donahue et al., 2011; Li et al., 2016) and $k_{OH}$ (Donahue et al., 2013); no methods for
estimation of HLC have been published.

Formula-based estimation of physicochemical parameters is necessarily less exact that structure-based estimation, as it has
less information available as an input (i.e., lack of structure). However, it has not been previously shown the extent to which
this lack of information degrades parameter estimation. If, for example, the uncertainty in parameter estimation is significantly
larger than differences caused by structure, there would be no significant loss in accuracy caused by not knowing the structure.
It is therefore an important but unanswered question to determine to what extent isomers differ in their parameters, and how
this compares to precision in parameter estimation. Addressing this issue would provide an understanding of the degree to
which it is relevant to know the structure of a molecule when estimating a given parameter. It is important to note that
application of SARs frequently include extrapolation beyond well-constrained laboratory data, which may decrease their
accuracy. Formula-based estimations are typically built off of these existing SARs, so inherently include their limitations and
biases. It is consequently less informative to discuss the accuracy of a formula-based estimation, which is driven in large part
by the underlying SAR(s) and for which experimental data is limited, but rather the precision of such a method, i.e., the ability
to recreate a structure-based estimate using only its molecular formula.

Given the large number of available methods, selection of a method for the estimation of a physicochemical parameter is non-
trivial, and researchers are left navigating a complex issue without obvious best practices. Selection of one method over another
is frequently an issue of convenience or familiarity, often with little consideration of the accuracy of a method, which may
itself be poorly constrained due to a lack of experimental data for atmospherically-relevant compounds. The range of choices
is further complicated by the fact that many methods have multiple publicly available implementations (e.g., online interfaces),
which we show in this work may disagree for a significant fraction of compounds. In an effort to understand the current
landscape, we examine here some widely used methods for the estimation of three critical physicochemical parameters: vapor
pressure, Henry's law constant (HLC), and $k_{OH}$. We combine these widely-used methods for estimation of these parameters to
answer several questions:

1.  How different are the various methods available for both structure-based and formula-based estimation of vapor
    pressure, Henry's law constants, and gas-phase OH reaction rates?
2.  Does knowing the structure of a molecule improve the estimation of its physicochemical parameters? *That is, are
    differences in physicochemical parameters between isomers sufficiently large to outweigh uncertainty in their
    estimation?*
3.  How much additional uncertainty is introduced in parameter estimation when structural information is unavailable?



## 2 Methods

To answer the questions posed above, physicochemical parameters were estimated for approximately 38,000 atmospherically-relevant species representing approximately 1,200 formulas. Parameters were estimated using a large number of methods currently in widespread use by the atmospheric chemistry scientific community. Differences between structure-based estimation methods for an individual compound were compared to differences between isomers of a formula for a given method. These were further compared to parameters estimated using formula-based methods. Details of species generation

and parameter estimation are provided below. Throughout the manuscript, notation used to describe derived quantities about a property, x, estimated by a structure-based estimation method (i.e., SAR), $m$, include:

- $\Delta x$ = difference in x between two isomers
- $\langle\Delta x\rangle_{formula}$ = average difference in x between all isomer pairs for a given formula
- $\Delta m_x$ = difference in x for a given species as estimated by two different SARs

- $\langle\Delta m_x\rangle$ = average difference in x between all SAR pairs for a given species
- $\bar{x}$ = average x of a species, estimated using all SARs
- $\bar{x}_{formula}$ = average x of a formula, estimated using all SARs for all isomers

Properties studied include: pure component subcooled liquid vapor pressure, $p$, in units of log(atm); Henry's law constant, $HLC$ or $H$, in units of log(M/atm), and gas-phase OH reaction rate, $k_{OH}$ or $k$, in units of $cm^3$/molec-s.

## 2.1 Generation of atmospherically relevant molecular structures

Atmospherically relevant species were generated using the simulated oxidation of precursor hydrocarbons. Three hydrocarbons – α-pinene, decane, and toluene – were selected to represent different chemical classes common in the atmosphere (cyclic alkene, saturated alkane, and aromatic, respectively) as well as different expected emissions sources. The gas-phase oxidation mechanism for these hydrocarbons were generated using the Generator of Explicit Chemistry and Kinetics of Organics in the

Atmosphere (GECKO-A). GECKO-A is a computer program designed to automatically generate the complete mechanism involved in the oxidation of a broad range of atmospherically important hydrocarbons. The tool generates chemical mechanisms according to a prescribed protocol, providing reaction rates based on experimental and theoretical data and SARs. The protocol implemented in GECKO-A is described by Aumont et al. (2005), with the chemistry updates given in Lannuque et al. (2018). The purpose of the study being to explore the properties of isomer distributions, oxidation was explicitly

considered up to the 5th generation and no lumping was here performed using surrogate species during the generation process. To limit the size of the mechanism, gas-phase chemistry for species having a vapor pressure below $10^{-13}$ atm was not generated, those species being expected to partition almost exclusively in the condensed phases under typical atmospheric conditions (e.g., Valorso et al., 2011). The numbers of species generated are $2.0\times10^5$, $5.5\times10^5$ and $7.5\times10^5$ for the decane, toluene and α-



pinene mechanisms, respectively. Non radical species are considered in both the gas and particle phase. Condensed phase
reactions are not considered in this model configuration.

Simulations are performed in a box model using conditions roughly representative of average continental atmospheric
conditions (Lannuque et al., 2018). In these runs, temperature is fixed at 298 K, photolysis frequencies are computed for mid-
latitude and for a solar zenith angle of 45° using the TUV model (Madronich and Flocke, 1999), the relative humidity is set to
70%. Mixing ratios are prescribed for methane (1750 ppb), CO (120 ppb), HCHO (2 ppb), $NO_x$ (500 ppt), $O_3$ (40 ppb).
Furthermore, a proxy species is introduced to include the influence of non-methane volatile organic compound oxidation on
the HOx and NOx cycles. First order loss rate of OH with respect to that proxy is set to 6 $s^{-1}$ and leads to the formation of a
surrogate peroxy radical, with a chemistry assumed to be similar to $CH_3O_2$. To allow gas/particle partitioning, a preexisting
mass concentration of organic particle is assumed and set to 10 $\mu g/m^3$. This condensed phase is assumed to behave as a well-
mixed ideal organic phase made of non-volatile organic matter. Finally, the parent hydrocarbon initial mixing ratio is set to an
arbitrary value of 10 ppt carbon, a value low enough to not modify substantially the prescribed buffered conditions. Time
integration of the mechanisms is performed for 5 days. These simulations served primarily to generate various species
representative of the molecular structures expected in typical ambient atmospheres under both high and low NOx conditions.
The analysis performed in this study is not sensitive to the exact oxidation conditions, as described below.

The number of species considered in the GECKO-A mechanisms is excessively large and a threshold was set in this work to
perform the analysis. The species representing the approximately 200 most abundant molecular formulas in each the gas and
particle phase were analyzed for each oxidation system. "Abundance" is considered here as the summed concentration across
the modeled period. Separately considering the abundance of gas- and particle-phase compounds ensures a dataset spanning
the atmospherically-relevant range of properties. Some of the same formulas may be abundant in both the gas- and particle-
phase components of a given oxidation system, but a given formula may be comprised of a different set of isomers or the same
isomers in different proportions. A total of 1193 formulas were consequently included in this analysis, roughly evenly split
between the three oxidation systems, as well as between gas- and particle-phase components.

For each formula, isomers were included in this analysis if they accounted for at least 0.1% of the abundance of each formula.
This threshold was selected to maximize statistical robustness by providing a large dataset, while minimizing the impact of
species expect to be produced at negligibly small concentrations. Selection of higher thresholds (e.g. 1%, 10%) were
investigated but not observed to significantly change the results of this work. In order to prevent this analyses from being too
strongly impacted by the specific chemistry of the model, isomers were not weighted by their abundance in any of the analyses
below; rather isomers were included with equal weight so long as they exceeded the 0.1% threshold. A total of 38,594 species
exceeded the 0.1% threshold in at least one oxidation system and phase. Each formula may include a variable number of



isomers, so compounds are not equally distributed between oxidation systems: 5% a-pinene gas-phase components, 6% decane gas, 10% toluene gas, 20% a-pinene particle, 16% decane particle, and 43% toluene particle. From this distribution it is apparent that in general, the model predicts particle-phase formulas to contain three to four times as many isomers as gas-phase formulas, and toluene oxidation produces twice as many isomers per formulas as the other two systems studied. Due to these differences, the six datasets are discussed separately where relevant throughout this work. Furthermore, species that have

both a gas- and particle-phase component exceeding the 0.1% threshold (N=3241 species) are included in both systems when gas- and particle-phase compounds are analyzed or discussed separately.

Each compound is described by a SMILES string from which physicochemical properties could be estimated computationally. Most structure-based estimation methods involve a two-step process in which the SMILES notation is parsed into the chemical functional groups relevant to the method, then the impact of each group is combined. All structure-based estimation in this

work was executed through publicly available online tools that performed both the parsing of the SMILES string as well as the computation of the properties, as described below.

## 2.2  Structure-based estimation of vapor pressure

### 2.2.1 SIMPOL

SIMPOL is a structure-activity relationship in which the subcooled liquid vapor pressure contributions of individual chemical functional groups are summed to generate a subcooled pure liquid vapor pressure (Pankow and Asher, 2008). No second-order interaction terms are included to account for neighboring functional groups. Two implementations of SIMPOL are publicly available: the GECKO-A online interface (http://geckoa.lisa.u-pec.fr/), and the Python package APRL Substructure Search Program, developed and made publicly available by Dr. S. Takahama (Ruggeri and Takahama, 2016). As of the time of

publication, the GECKO-A online interface does not accept standard SMILES strings, requiring instead a modified notation that uses explicit hydrogens and a few other differences, making its widespread use somewhat more difficult. Both implementations of SIMPOL were compared as part of this work. While small differences are expected due to uncertainty in parsing SMILES notation and ambiguity in chemical functional group assignment, vapor pressures estimated by SIMPOL should ideally be nearly identical between implementations. In the case of decane and a-pinene oxidation products, these

implementations were in excellent agreement (Figure S1). However, significant differences were observed in their estimations of toluene oxidation products having complex molecular structures. To understand the differences observed for toluene oxidation products, SIMPOL was implemented manually for a random set of compounds that were observed to not agree, with results in Table S1. While some differences may be attributable to real errors in implementations, a larger uncertainty appears to be associated with needing to extrapolate beyond the functional groups identified within the SIMPOL SAR. For example,

SIMPOL does not include the α-carbonyl peroxide (-C(=O)-O-O-R) functional group; while a peroxide group is included,



carbonyls are included only as ketones and aldehydes, neither of which is an accurate description of this case. APRL treats this group as a peroxide, with no contribution from the carbonyl group, while GECKO-A treats this group as an ester-ether; little or no data exists to determine which approach is more accurate. This example points to a systematic limitation of SARs, and the inherent potential differences between implementations for complex atmospheric oxidation products.

In the case of SIMPOL, manual investigation suggests that most differences between implementations could be traced to differences in the interpretation or extrapolation of the SAR for functional groups outside the prescribed bounds. Neither implementation was found to be clearly more suitable or faithful to the published SAR. The GECKO-A implementation of SIMPOL was used in this work because the online interface of GECKO-A provides a logistical benefit by implementing this method alongside multiple other structure-based parameter estimations. Results in this work are found to be relatively
insensitive to the choice of implementation as they are nearly identical for decane and α-pinene oxidation products.

### 2.2.2 EVAPORATION

EVAPORATION is a structure-activity relationship for the estimation of subcooled liquid vapor pressure that includes vapor pressure contributions of individual chemical functional groups, as well as terms to account for interactions between neighboring groups (Compernolle et al., 2011). Currently, this method lacks terms to describe several less abundant but
nevertheless atmospherically-relevant functional groups, including $-NO_2$ and $-C(=O)ONO_2$. For the purpose of this analysis, these groups were replaced by $-ONO_2$ and $-C(=O)OONO_2$ respectively, which are predicted to have similar impacts on vapor pressure based on SIMPOL and EPI. EVAPORATION currently also lacks a treatment of aromaticity, but this limitation has little impact on this dataset. Though toluene is aromatic, oxidation quickly breaks its aromaticity and fewer than 200 oxidation products contained aromatic carbon; aromatic carbons were replaced with aliphatic carbons for these compounds, which is
expected to introduce bias of approximately half an order of magnitude for this small subset of compounds.

Two implementations of the EVAPORATION method are publicly-available as online resources. A direct online interface is available through the Royal Belgian Institute for Space Aeronomy (hereafter referred to as "IASB", found at https://tropo.aeronomie.be), the institution at which the SAR was developed. A separate implementation is available as part of the UManSysProp package for the estimation of a wide range of physicochemical and system parameters, developed and
published by researchers at the University of Manchester (Topping et al., 2016). UManSysProp is available both as a standalone Python package, and an online interface at http://umansysprop.seaes.manchester.ac.uk.

Both the IASB and UManSysProp implementations of EVAPORATION were compared as part of this work in order to ensure that inclusion of this estimation method in this work is as faithful as possible to the published SAR. Though the comparison of these implementations shown in Figure S2 fell generally along a one-to-one line as expected, some significant differences
were observed. Vapor pressures estimated for decane oxidation products were almost always nearly identical, but oxidation products of α-pinene differed by approximately an order of magnitude for a large fraction of the tested compounds and toluene





oxidation products differed significantly and variably for a substantial majority of compounds. To assess these differences, the EVAPORATION SAR was tested manually for a small set of compounds that differed between implementations. Values manually computed were found in most cases to be in reasonable agreement with the IASB implementation, but frequently

differed from the UManSysProp implementation (Table S2). Not all differences in methods could be obviously explained by extrapolation beyond prescribed functional groups, but these differences nevertheless highlight the difficulties encountered in implementing a given SAR for highly diverse and complex molecular structure. This work relies on the IASB implementation for estimation of vapor pressures by the EVAPORATION method based on its agreement with manual implementation and the fact that this implementation is provided by the institution at which the SAR was developed. We note that the open-source

nature of the UManSysProp package allows a user to understand and/or modify its source code, so future updates may impact these comparisons, but no attempt was made in this work to reconcile the two methods.

### 2.2.3 Nannoolal

Nannoolal and co-workers developed a group contribution method for the prediction of vapor pressure given the structure and boiling point of a molecule (Nannoolal et al., 2008). This method includes a substantially larger number of groups than either

SIMPOL or EVAPORATION, encompassing a broader range of compounds including inorganic groups, and includes second-order terms to account for interactions between neighboring groups. Boiling point can in turn be estimated from molecular structure using a group contribution method developed by Nannoolal and co-workers (Nannoolal et al., 2004). "Nannoolal" in this work refers to the estimation method using both the vapor pressure and boiling point group contribution methods developed by Nannoolal et al. Two implementations of Nannoolal are available through online interfaces, specifically using the GECKO-

A interface and the UManSysProp package. Some differences were observed between these implementations (Figure S3), similarly in scope and scale to the EVAPORATION comparison above. It is clear from the comparisons of Nannoolal and EVAPORATION implementations that estimation of vapor pressures for toluene oxidation products pose unique complexities. Due to the general similarity between implementations for the non-aromatic precursors and the use of the Nannoolal SAR as the default estimation method in the GECKO-A model itself, no further examination of the implementation in the two tools

was undertaken. In this work, Nannoolal refers to the GECKO-A implementation of this method.

### 2.2.4 Myrdal and Yalkowsky

The vapor pressure estimation method developed by Myrdal and Yalkowsky consists of a group-contribution correction to a previous semi-empirical estimation method that relied only on boiling and melting points, as well as estimations of the entropy of boiling, entropy of melting, and heat capacity change upon boiling (Myrdal and Yalkowsky, 1997). In this modification, a

small number of groups (fewer than a dozen) and molecular properties (e.g. rotational symmetry) are considered for their impacts to these three estimated physicochemical properties. For calculation of subcooled liquid vapor pressures, the terms considering temperatures and entropies of melting can be ignored. Consequently, vapor pressure estimation by the Myrdal and Yalkowsky for this work depends only on molecular structure and boiling points.


This work relies on the UManSysProp implementation of the Myrdal and Yalkowsky method, which allows estimation of
boiling point by any of several methods. For this work, boiling points were estimated using the Nannoolal estimation technique
(Nannoolal et al., 2004). Another implementation is available through the GECKO-A interface using the Joback and Reid
boiling point group contribution estimation technique (Joback, 1984; Reid et al., 1987), with some modifications as described
by Camredon et al. (2006). These implementations are not compared in this work because this SAR has been shown previously
to be less accurate (Barley and McFiggans, 2010), and so is not included in most of the analyses in this work.

**2.2.5 EPI**

The U.S. Environmental Protection Agency makes available the Estimation Programs Interface Suite™ for the estimation of
environmentally relevant parameters (US Environment Protection Agency, 2019), which includes a module ("MPBPVP") for
the estimation of vapor pressures and subcooled liquid vapor pressures using SMILES strings as inputs. This module uses the
"Modified Grain Method" which estimates vapor pressure based on a near-unity structural factor and an estimated boiling
point. Boiling point is in turn estimated using the Stein and Brown group contribution method (Stein and Brown, 1994), an
extension of the Joback and Reid method (Joback, 1984; Reid et al., 1987). This approach includes group contributions for a
wide variety of molecular structures, including a wide range of inorganic components. Estimation of vapor pressures by the
EPI Suite is perhaps most common for estimating small numbers of vapor pressures due to its readily available implementation,
though it has somewhat higher error than some other methods (e.g., Nannoolal) when compared against experimental data
(Barley and McFiggans, 2010, wherein the method SB/BK closely approximates the EPI method).

**2.3 Structure-based estimation of Henry's law constant**

Two structure-based methods were considered in this work for the estimation of Henry's Law Constants (HLCs). One method
used here is "HWINb", the bond contribution method implemented by the HENRYWIN module of the EPI Suite (US
Environment Protection Agency, 2019). This method is similar to a group contribution method, but instead of using groups,
individual bonds are considered with correction factors for different chemical classes (Hine and Mookerjee, 1975; Meylan and
Howard, 1991). The other method used here is "GROMHE" (GROup contribution Method for Henry's law Estimate), a group
contribution method that also includes a group contribution term for the effect of hydration of carbonyls (Raventos-Duran et
al., 2010). GROMHE is the HLC estimation method used by GECKO-A, which is the implementation used in this work.
Previous work has suggested GROMHE to be more accurate than HWINb, but this conclusion was based on a relatively small
amount of experimental data (<500 compounds) with relatively low HLCs (Raventos-Duran et al., 2010). We consequently do
not assume the accuracy of one method over another and instead assume the variability between methods is due to uncertainty
in structure-based estimation of HLC. The GECKO-A implementation of GROMHE sets a threshold of $10^{18}$ M/atm for HLC,
capping all higher estimations at this value; consequently species estimated by either GROMHE or HWINb as having HLC at
or above this threshold are excluded from all analyses in this work.



### 2.4 Structure-based estimation of $k_{OH}$

Two structure-based methods were considered in this work for the estimation of $k_{OH}$. Perhaps the most common method is that developed by Kwok and Atkinson, a group contribution method that includes additive terms for hydrogen abstraction from or radical addition to individual atoms or bonds (Kwok and Atkinson, 1995). An additional second-order term accounts for substituent effects on each atom. The implementation of this method used here is the AOPWIN module of the EPI Suite (Meylan and Howard, 1993; US Environment Protection Agency, 2019) . The other method used here is the group contribution method of Jenkin et al., which functions similarly to Kwok and Atkinson approach but with updated and extended coefficients (Jenkin et al., 2018a, 2018b). Jenkin et al. is the $k_{OH}$ estimation method used by GECKO-A, which is the implementation used in this work, and which has an available interface through the GECKO-A online interface.

### 2.5 Formula-based estimation of vapor pressure

### 2.5.1 Daumit et al.

Daumit and co-workers use a basic set of assumptions about the structures of atmospheric components to apply the SIMPOL estimation method in the absence of molecular structure (Daumit et al., 2013). Essentially, all oxygen atoms in a molecule are apportioned between hydroxyl and carbonyl groups based on the degree of unsaturation calculated from the H/C and O/C ratios. To accurately calculate degrees of unsaturation, an assumption must be made about the number of rings present in the molecule. We assume there to be no rings, as this is consistent with the majority of compounds in this dataset, but the need to make this assumption represents a general source of uncertainty in the Daumit et al. method. While Daumit et al. do not explicitly treat nitrogen, they note that the nitrate group is expected in SIMPOL to have a similar impact as the hydroxyl group. To explicitly extend this method to nitrogen, we make the assumption that nitrogen is predominantly present as nitrate groups and each nitrate group is treated as equivalent to a hydroxyl group. For every three oxygen atoms present in the formula, two oxygen atoms and one nitrogen atom is removed until all nitrogen has been removed. The resulting formula, in which all possible $NO_3$ groups have been formulaically converted to OH, are treated as per Daumit et al. As an example, the formula $C_8H_{15}O_6N$, interpreted as containing one nitrate group, one carbonyl, and two hydroxyl groups, would be treated as $C_8H_{16}O_4$, interpreted as containing one carbonyl, and three hydroxyl groups. In environments in which nitrogen is present in forms other than nitrate, Daumit et al. lacks an explicit mechanism for considering nitrogen. An additional limitation of this approach is that while certain groups can be approximated as a combination of carbonyl and hydroxyl oxygens, others may be poorly described in this way. For example, the vapor pressure contribution of a carboxylic acid is estimated to be similar to that of a ketone or an aldehyde plus a hydroxyl group, but a hydroperoxide has a substantially lower impact than that of two hydroxyl groups.





### 2.5.2 Modified Li et al. ("molecular corridors")

The formula-based approach for the estimation of vapor pressures developed by Li et al. as part of their work on "molecular corridors" uses empirical coefficients to quantify the impact of each atom on vapor pressure, with a minor term for interactions between carbon and oxygen (Li et al., 2016). Formulas are first categorized by their component elements, with a separate set of coefficients for e.g., CHO formulas vs. CHON formulas. This method was developed by multi-linear regressions against a training set of vapor pressures estimated by the EPI Suite, and is thus to some extent limited to the compound classes on which

it was trained and can only be as accurate as the EPI estimation method. Most notably, despite the relative prevalence of organic nitrates (R-ONO$_2$) in the atmosphere (Lee et al., 2016), few such compounds exist in the CHON training set used by Li et al. Of the 13,628 CHON compounds used to build the relationship, only 9 (0.07%) are organic nitrates and 750 (5.5%) are organic nitro compounds, which have a similar impact on vapor pressure; all other included compounds represent amines, amides, amino acids, and other groups that contain C-N bonds, which are expected to have a very different impact on vapor

pressure. Consequently, application of the Li et al. formula-based estimation technique to compounds containing nitrates is expected to be significantly biased. We test this hypothesis here in order to more accurately apply this method to the dataset.

       Comparison of vapor pressures estimated by Li et al. to vapor pressures estimated for the same compound using structure-based methods (Figure S4) demonstrates significant biases that increase with the number of nitrogen atoms, which in this dataset are almost wholly contained in nitrate, nitro, and peroxynitrate groups. To address this bias we propose two similar

possible approaches based on the observation that a nitrate group (NO$_3$) has a similar impact on vapor pressure as a hydroxyl group (OH), and thus each nitrogen has the effect of canceling the effect of two oxygen atoms. Either the nitrogen coefficient for CHON formulas can be forced to equal twice the negative of the oxygen atom ($b_N = -2*b_O$), or the formula used to estimate vapor pressure can be amended to convert all potential nitrate groups into hydroxyl groups as described in the implementation of Daumit et al. Both approaches are shown in Figure S4 to similarly remove the nitrogen-dependent bias, and are generally

equivalent in this dataset. In mixed environments in which oxygenated amines and organonitrates may co-exist, formulaically converting nitrate groups to hydroxyl groups may be preferred in order to more accurately treat nitrogen in excess of potential nitrate groups (i.e. in cases where the number of nitrogens is greater than the number of sets of 3 oxygens). However, given the nitrate-dominate nature of this dataset, for simplicity we use a Modified Li et al. method in which $b_N = -2*b_O$.

### 2.5.3 Donahue et al.

A relatively simple formula-based estimation method is provided by Donahue et al. (2011), relying only on carbon and oxygen number. This method represents a general relationship based on average expected trends in the structures of atmospheric components. It cannot be easily extended to nitrogen-containing formulas, so they are excluded from analyses using this approach in the present work.



### 2.6 Formula-based estimation of Henry's law constant

To the best of our knowledge, no explicit method for formula-based estimation of Henry's law constant (HLC) has been published. However, explicit modeling of gas phase oxidation has previously shown a relationship between HLC and vapor pressure for organic species of atmospheric interest (Hodzic et al., 2014; Lannuque et al., 2018). Given the previously demonstrated feasibility of formula-based estimation of vapor pressure, this suggests formula-based estimation of HLC is possible, which we demonstrate later in this work.

### 375 2.7 Formula-based estimation of $k_{OH}$

In separate work from their formula-based vapor pressure estimation, Donahue and co-workers (Donahue et al., 2013) have developed a formula-based approach to the estimation of gas-phase OH reaction rates ($k_{OH}$). The equation they provide is roughly based on the observations that as carbon number increases, available hydrogens for OH abstraction also increase, and as oxygen number increases, hydrogens become easier to abstract but there is a decrease in the number of abstractable 380 hydrogens. Donahue et al. recognize it only as a rough approximation and not a particularly robust estimation method, a conclusion consistent with results shown below.

### 2.8 Summary

Given the large number of methods employed in this work, we summarize below alongside the notation hereafter used in this work.

*Structure-based estimation of vapor pressure:*

- "SIMPOL" – calculated from SIMPOL as implemented by GECKO-A
- "EVAPORATION" – calculated from EVAPORATION as implemented by the Royal Belgian Institute for Space Aeronomy ("IASB")
- "Nannoolal" – calculated based on Nannoolal et al. (2008) using boiling points estimated by Nannoolal et al. (2004), 390 as implemented by GECKO-A
- "Myrdal and Yalkowsky" – calculated based on Myrdal and Yalkowsky (1997) using boiling points estimated by Nannoolal et al. (2004), as implemented by the UManSysProp Python package
- "EPI" – calculated by the EPI Suite, an implementation of the Modified Grain Method using boiling points estimated by Stein and Brown (1994).

*Structure-based estimation of Henry's Law Constant:*

- "HWINb" – calculated by the EPI Suite, using the bond contribution method of the HENRYWIN module





- "GROMHE" – calculated with the GROMHE group contribution method, as implemented by GECKO-A

*Structure-based estimation of $k_{OH}$:*

- "Kwok and Atkinson" – calculated based on Kwok and Atkinson (1994) method, as implemented by the AOPWIN
module of the EPI suite
- "Jenkin" – calculated based on Jenkin et al. (2018a, 2018b), as implemented by GECKO-A

*Formula-based estimation of vapor pressure:*

- "Daumit" – calculated based on Daumit et al. (2013), with consideration for nitrates
- "Modified Li" – calculated based on Li et al. (2016), with modified nitrogen coefficient
- "Donahue" – calculated based on Donahue et al. (2011), not used for nitrogen-containing formulas

*Formula-based estimation of Henry's Law Constant:*

   None previously published

*Formula-based estimation of $k_{OH}$:*

- "Donahue" – calculated based on Donahue et al. (2013), not used for nitrogen-containing formulas


## 3 Results

### 3.1 Isomer differences for vapor pressures

A primary objective of this work is to understand typical differences in vapor pressures between isomers. We evaluate these differences here by calculating the average difference in the estimated vapor pressure of any two isomers of a given formula.
For each formula containing $n$ isomers, $(n*(n-1)/2)$ distinct pairs of isomers can be counted. For each possible pair of isomers $i$ and $j$, the absolute difference in the estimated log vapor pressure is computed as $\Delta p = |\log(p_i) – \log(p_j)|$. The average difference in vapor pressure among isomers of a given formula (denoted $<\Delta p>_{formula}$ hereafter) is then computed as the average of the $\Delta p$ obtained for all pairs of a given formula. For all five structure-based vapor pressure estimation methods included in this work, $<\Delta p>_{formula}$ is relatively evenly distributed between 0 and 2 log units (Figure 1a). The overall average of $<\Delta p>_{formula}$ is between
0.8 and 1.0 log units across all five estimation methods, indicating that the central tendency is for two isomers to differ by approximately one log unit in vapor pressure. The distribution of $<\Delta p>_{formula}$ depends on the oxidation system studied, as is



clear from the breakdown of distributions by precursor and phase shown for Nannoolal in Figure 1b; the trends observed for Nannoolal are generally representative of the other four methods, shown as Figures S6 and S7. Vapor pressures of isomers are more similar for decane oxidation products ($<\Delta p>_{formula} \approx 0.5$ log units), and less similar for toluene oxidation products

($<\Delta p>_{formula} \sim 1.5$ log unit), with a-pinene oxidation products in between ($<\Delta p>_{formula} \approx 1$ log unit). Phase of the compound, which serves in large part as a proxy for volatility (i.e. particle-phase components have lower vapor pressures), also has some impact, with somewhat higher $<\Delta p>_{formula}$ for formulas abundant in the particle phase. Note that the components are distinguished as gas- and particle-phase based on their abundance in either phase – a minor fraction of species is represented in both datasets.

The $<\Delta p>_{formula}$ metric obscures some of the larger individual differences between pair of isomers. The complete cumulative

frequency distribution of $\Delta p$ is shown in Figure 1c for all isomerpairs. While 50% of $\Delta p$ values differ by less than 1 log unit, a long tail indicates that in many cases isomers may differ by up to around 3 log units (or, rarely, 4 or 5 log units). These trends are true across all five tested estimation methods. The various oxidation systems (Figure 1d) vary in their $\Delta p$ cumulative frequency distribution in qualitatively similar ways as their distributions of $<\Delta p>_{formula}$: toluene oxidation isomers differ substantially more in their vapor pressures than the isomers in other systems, and gas-phase isomers are slightly less variable

in their vapor pressures than particle-phase isomers.

It is consequently difficult to provide a single number to characterize the typical $<\Delta p>_{formula}$ values due to the wide distribution, variabilities between systems, and differences between methods. However, it is a reasonable overall summary that vapor pressures of isomers differ by between 0.5 and 3 log units, with a central tendency of ~1 log unit. Estimation methods typically agree about the range of $<\Delta p>_{formula}$, but it is sensitive to the oxidation system being studied.






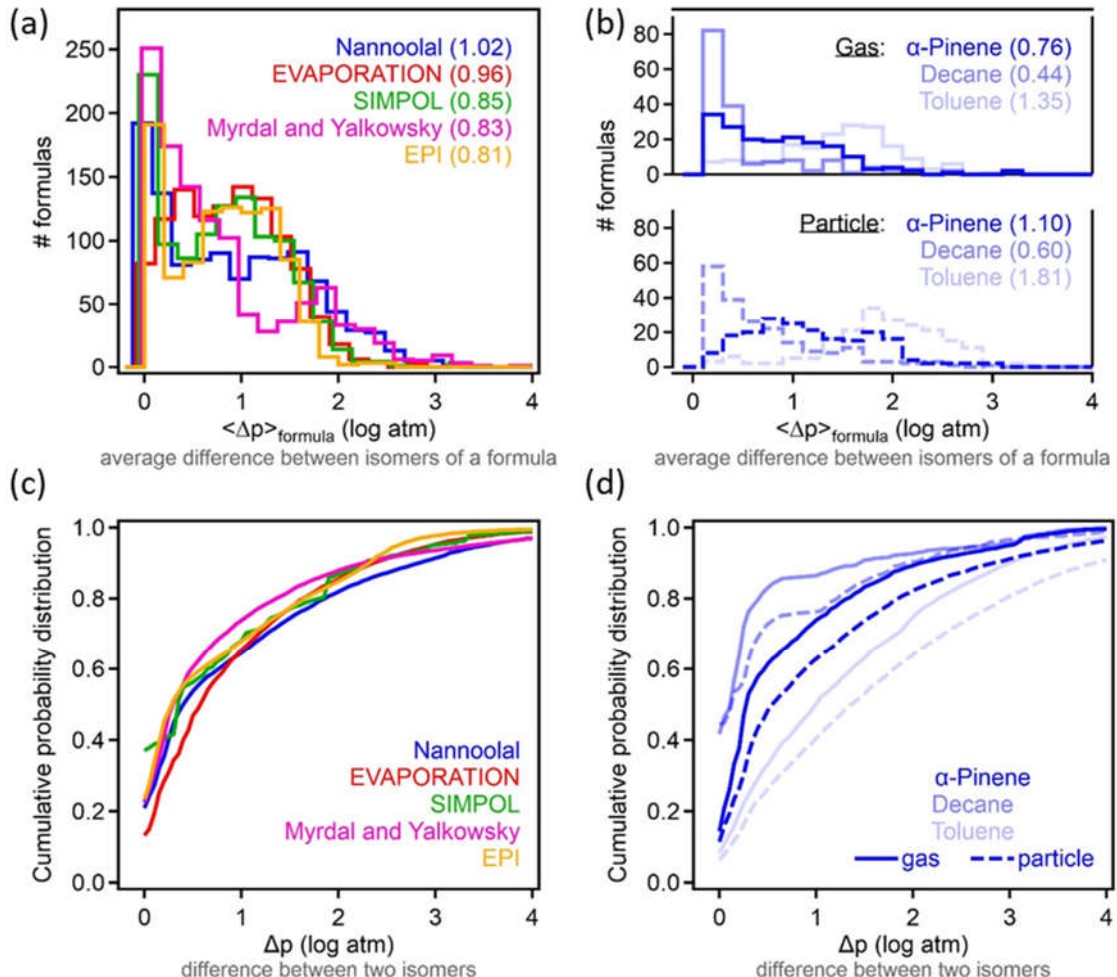

**Figure 1. Differences in vapor pressure between isomers. (a) Distribution of $<\Delta p>_{formula}$, the average difference between vapor pressures of isomers of a given formula for the five structure-based estimation methods examined, with (b) the same distribution broken out by oxidation system for the Nannoolal method. Average values of each distribution are provided in parentheses. (c) Cumulative probability distribution of $\Delta p$, the difference between any two isomers of a given formula for the five structure-based estimation methods examined, with (d) the same distribution broken out by oxidation system for the Nannoolal method. The other four methods are shown in Figures S6-S7.**

Though one log unit (a factor of 10), is a substantial difference in vapor pressures, it must be placed in the context of our ability to estimate the parameter. In other words, if estimation methods differ by more than this for a given species, details of the molecular structure are less important than which estimation method is used, so knowing the molecular structure would not substantively improve the estimate. In the Supplementary Information we compare EPI, Myrdal and Yalkowski, SIMPOL and EVAPORATION versus Nannoolal estimation methods both as scatter plots (Figure S8), and histograms of the difference between two methods (Figure S9). The Myrdal and Yalkowsky (using Nannoolal boiling point estimation) and EPI methods estimate substantially higher vapor pressures for low-volatility compounds than the other three methods, consistent with



previous work (Compernolle et al., 2011). This trend is in agreement with previous work that has shown overestimation of
vapor pressures, particularly at lower vapor pressures, by the Myrdal and Yalkowsky method and the Stein and Brown method
upon which EPI is based (Barley and McFiggans, 2010). In turn, Nannoolal estimates somewhat lower vapor pressures than
SIMPOL and EVAPORATION for low-volatility compounds, but to a lesser extent. Similar trends between SIMPOL,
Nannoolal, EVAPORATION, and Myrdal and Yalkowsky have been previously shown for the oxidation products of α-pinene
(Compernolle et al., 2011; Valorso et al., 2011). There is no sufficiently large database of known vapor pressures to know
which of these methods is most accurate in these regions. We instead assume that the best available estimate for the vapor
pressure of a compound is the average of the SIMPOL, Nannoolal, and EVAPORATION estimates. This assumption is based
on the similarity of the SIMPOL, Nannoolal, and EVAPORATION methods in this and in previous work, previous work
demonstrating agreement between Nannoolal and experimental data (Barley and McFiggans, 2010), and the more recent
development of these SARs (2008, 2008, and 2011, respectively). The EPI and Myrdal and Yalkowsky methods are treated as
outliers based on their overestimation of low vapor pressures relative to experimental data (shown by Barley and McFiggan,
2010) and to other methods (Figure S8). By averaging the vapor pressures estimated for each species with Nannoolal, SIMPOL
and EVAPORATION methods, we mitigate any biases present in any one method. The average of these three methods provides
an average structure-based estimate for a given species, denoted here as $\bar{p}$.

To understand precision in structure-based estimation, we quantify the differences between methods in the predicted property
of a given species. For each species, the vapor pressure is estimated using the three selected methods above. We denote $\Delta m_p$
as the absolute difference in vapor pressure of a given species between any 2 methods $q$ and $r$ ( $\Delta m_p = |\log(m_{p,q}) - \log(m_{p,r})|$ )
and $<\Delta m_P>$ as the average value for the 3 possible combinations. The $<\Delta m_P>$ frequency distribution is shown in Figure 2a-b.
For gas-phase components, $\Delta m_p$ is within one log unit, with $<\Delta m_P>$ around 0.5 log units. This is in reasonable agreement with
reported uncertainties for each individual method. Estimation methods appear to have somewhat less skill for particle-phase
compounds. For these lower-volatility compounds, $<\Delta m_P>$ is around 1 log unit, with most compounds within 2 log units. Note
that for both gas- and particle-phase compounds, toluene oxidation products again tend to differ more in their estimated vapor
pressures. In other words, while isomers for this system have higher vapor pressure differences, models are also less reliable
at estimating this property; these facts may be related (high uncertainty in estimation may contribute to larger differences
between isomers).

The difference in the variability between estimates for gas- versus particle-phase components is primarily a function of
differences in volatility. This issue is qualitatively observed in the direct comparison between methods shown in Figure S8, in
which methods diverge at lower vapor pressure, but is examined more explicitly here. Figure 2c shows $<\Delta m_P>$ as a function
of average vapor pressure, $\bar{p}$, for all species considered here (red dots); averages (and standard deviations) of ten bins of equal
points each (deciles) are shown to make trends clear. At higher vapor pressures, differences between methods remain under 1
log unit, while this increases substantially at the lowest vapor pressures (and oxidation products decane and a-pinene always





have lower $\langle \Delta m_p \rangle$ than those from toluene). Increasing $\langle \Delta m_p \rangle$ at low volatility is not unexpected, as most estimation methods require extrapolation from relatively high vapor pressure data. Furthermore, volatility in this dataset acts in part as a proxy for functionality. All species studied are formed through the oxidation of three relatively volatile precursors, so decreases in volatility are generally caused by the addition of oxygenated functional groups. The decrease in vapor pressure caused by each functional group is of course uncertain, so methods diverge as number of functional groups increase and volatility decreases (Valorso et al., 2011).

As in our discussion of vapor pressure differences between isomers, it is difficult to provide a single number to characterize our skill in estimating vapor pressure from a molecular structure. It is a reasonable overall summary that higher vapor pressures can be estimated within 1 log unit, with a central tendency of ~0.5 log unit. This $\langle \Delta m_p \rangle$ range is somewhat smaller than typical differences between isomers, $\langle \Delta p \rangle_{formula}$. We estimate that the effect of isomers is 0.5-1.5 log units greater than the variability between estimation methods for high-to-moderate vapor pressures. At lower vapor pressures however, $\langle \Delta p \rangle_{formula}$ are not substantially larger than $\langle \Delta m_p \rangle$, so the impact of structure is less than variability in estimation methods. The transition vapor pressure below which differences between isomers are lost in the uncertainty of our methods does not (of course) occur at a single point, but can reasonably be considered to be in the range of $10^{-10}$ to $10^{-12}$ atm ($c^* \approx 10^{-2.5}$ to $10^{-0.5}$ μg/cm$^3$), where the average difference between methods, $\langle \Delta m_p \rangle$, is approximately equal to the average difference between isomers, $\langle \Delta p \rangle_{formula}$ (~1 log unit). This suggests that the difference in vapor pressures between isomers is likely relevant for estimating vapor pressures of semivolatile compounds – those that can partition back and forth between the gas and particle phases under typical atmospheric conditions (roughly $c^* \approx 10^{-0.5}$ to $10^{2.5}$ μg/cm$^3$ per Donahue et al. (2009, 2011).




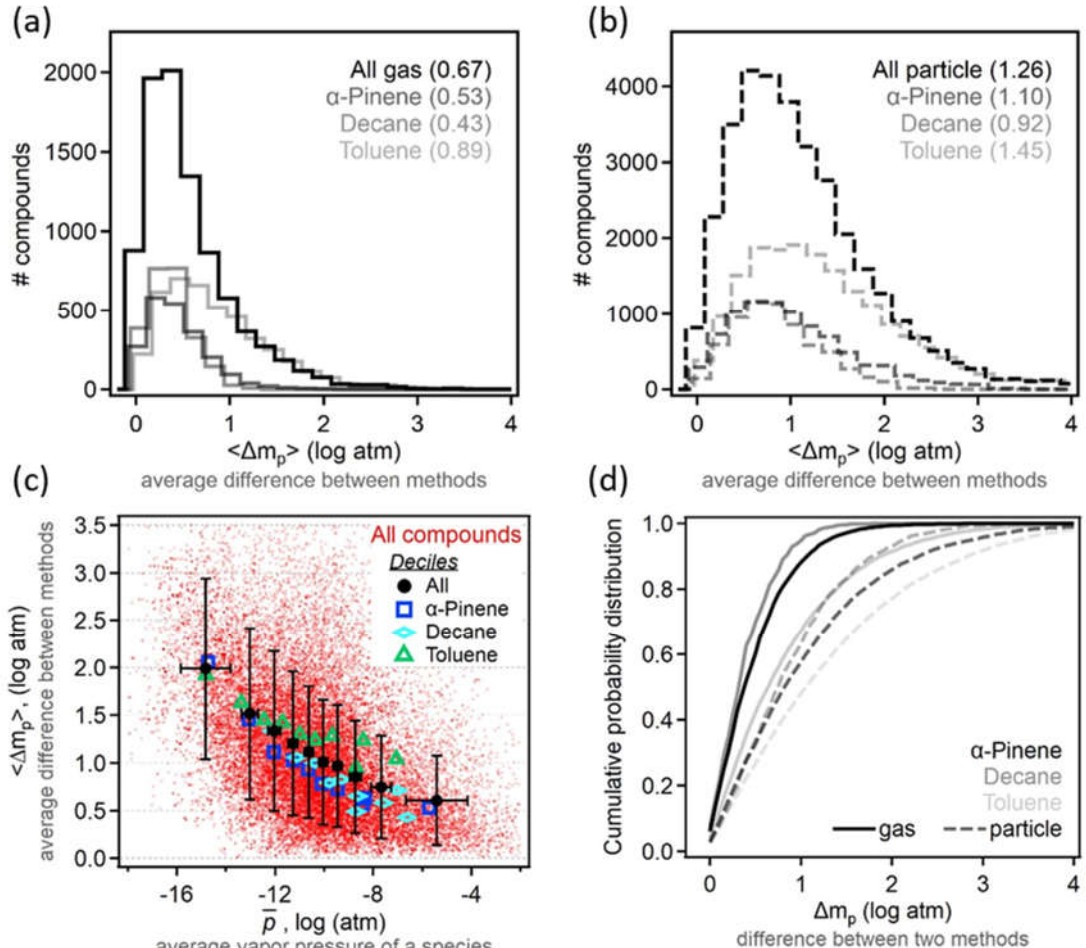

**Figure 2. Differences in vapor pressures between the Nannoolal, SIMPOL, and EVAPORATION estimation methods.**
**(a) Distribution of $\langle\Delta m_P\rangle$, the average difference between vapor pressures estimated for a given compound in the (a)**
**gas and (b) particle phase, which each oxidation system shown in a different lightness. Average values of each**
**distribution are provided in parentheses. (c) Distribution of $\langle\Delta m_P\rangle$ as a function of vapor pressure (as average vapor**
**pressure of a species, $\bar{p}$), broken out by oxidation system. Red dots are individual species, larger markers and error**
**bars are the average and standard deviation of deciles. (d) Cumulative probability distribution of $\Delta m_P$, the difference**
**between any two methods for given species.**

## 3.2 Estimation of vapor pressure by formulas

The above analysis indicates that isomers are sufficiently different between their vapor pressures that structure should be taken

into account when estimating this parameter. However, due to the increasing use of mass spectrometric instruments that

measure atmospheric constituents by their formulas with no accompanying structural information, there is an increasing need





to estimate vapor pressure and other parameters by formula only. Formula-based estimation will necessarily be more uncertain
as it relies on less information (i.e., lacks molecular structure), and this approach cannot be more precise that than the impacts
of structure on a property. A goal of this work is to assess the precision of current formula-based estimation approaches. For
each formula, an average vapor pressure of a formula (denoted $\bar{p}_{formula}$) is computed as the average $\bar{p}$ of all isomers of that
formula. $\bar{p}_{formula}$ therefore represent a "composite structure-based estimate" of the vapor pressure using the three structure-
based methods (i.e. SIMPOL, Nannoolal, EVAPORATION) and all isomers. Including all isomers and all methods in the
average of each formula provides the most direct possible comparison, mitigating bias introduced by any one structure-based
estimation method or uncertainties driven by any one isomer. The standard deviation of this average, $\sigma_P$, also provides an
estimate of the range of the vapor pressures that species of a given formula may have. This range represents the variability in
vapor pressure driven by differences in molecular structure, accounting for both differences between isomers and between
SARs, and thus provides an estimate of the maximum precision of an estimation method that ignores structure. Assuming an
approximately normal distribution, approximately two-thirds of species of a formula are expected to have a vapor pressure
within the range of $[\bar{p}_{formula} - \sigma_P, \bar{p}_{formula} + \sigma_P]$. The reliability of the three formula-based estimation methods (Daumit,
Modified Li, and Donahue) is assessed by comparing their estimated vapor pressure with $\bar{p}_{formula}$ (Figure 3). An unbiased
formula-based estimation would be expected to fall along a 1:1 line, with two-thirds of estimates falling within the expected
range of $[\bar{p}_{formula} - \sigma_P, \bar{p}_{formula} + \sigma_P]$.


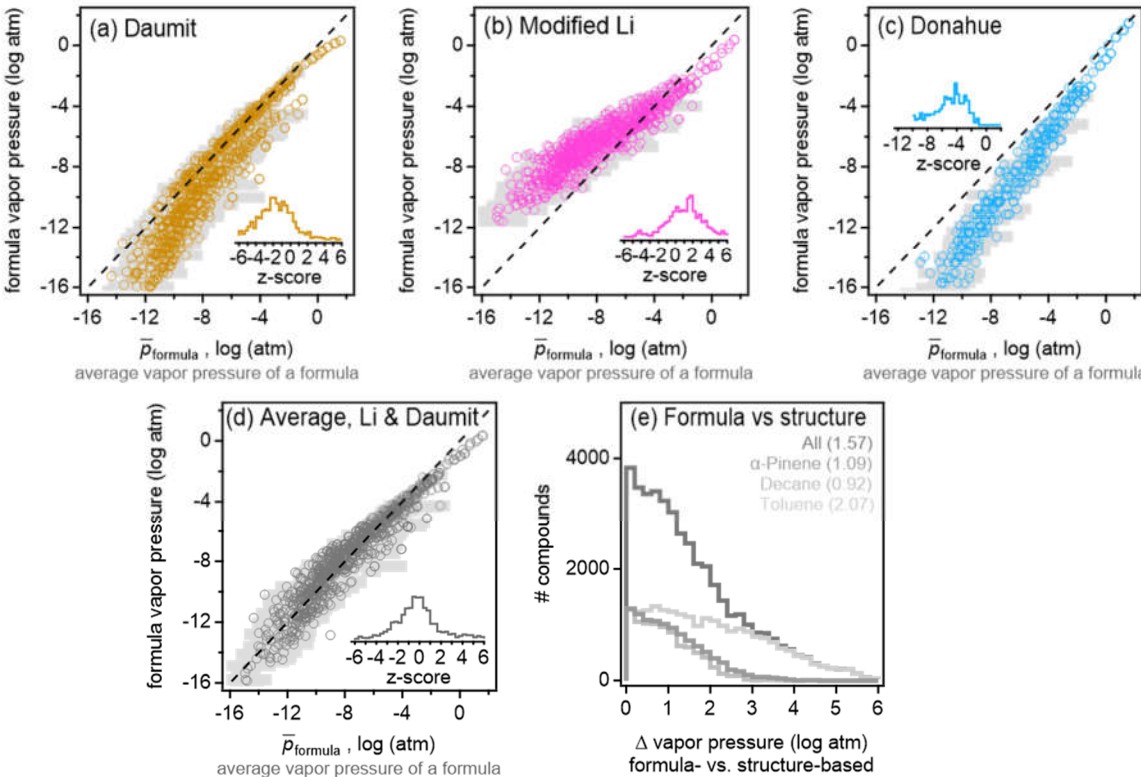

**Figure 3. Comparison between average vapor pressure of a formula $\overline{p}_{formula}$ (average of all methods and all isomers, see text) and the formula-based estimate using the (a) Daumit method, (b) Li method, modified to remove its bias for nitrates, (c) Donahue method and (d) Average of Daumit and modified Li methods. Each formula is represented as an open circle at $\overline{p}_{formula}$, with light-gray bars representing standard deviation of the average, $\sigma_P$, to indicate the approximate range. Insets are distributions of z-scores for each method, calculated as difference between formula-based method and $\overline{p}_{formula}$, relative to the standard deviation of $\overline{p}_{formula}$. (e) Distribution of error from applying the average Daumit-Li method to any given compound, with each oxidation system shown in a different lightness (gas and particle phases combined). Average values of each distribution are provided in parentheses.**

Biases and uncertainty in the three formula-based estimation techniques can be understood in the context of their development. All three methods demonstrate relatively high skill at estimating vapor pressures for more volatile components, where isomer differences are lower and structure-based estimation methods tend to agree due to better constraints. The formula-based methods diverge from each other and from the composite structure-based estimate at lower vapor pressures. The Daumit method (Figure 3a) tends to estimate lower vapor pressures than expected, which is predictable upon closer inspection of this method. Daumit treats all oxygen as a combination of hydroxyl and carbonyl groups, which is reasonable in some cases (e.g., carboxyl acids). In cases where this approximation does not hold, it is generally true that the decrease in vapor pressure caused by a functional group is less than sum of its component oxygens. For example, peroxides have relatively little impact on vapor pressure but will be treated as two hydroxyl groups as discussed above in section 2.5.1. As the number of groups increases,




vapor pressure decreases "faster" than it should, leading to a low bias in the Daumit method. Conversely, the Li method
(implemented here with a modified nitrogen coefficient) is based on vapor pressures calculated by the EPI method, which
tends to estimate higher vapor pressures for low-volatility species (Figure S8). Consequently, the Li method follows the same
trend, estimating higher-than-expected vapor pressures at low vapor pressures (Figure 3b). The Donahue method (Figure 3c)
roughly follows but exceeds the biases of the Daumit method as it is based on more simple assumptions about molecular
structure (and cannot treat nitrogen-containing components). In general, the formula-based estimations from all three methods
fall well outside the range of $\bar{p}_{formula}$. Distributions of z-scores are shown as insets, calculated as the difference between the
formula-based estimate and $\bar{p}_{formula}$, relative to the standard deviation of $\bar{p}_{formula}$, i.e., z-score = $(p-\bar{p}_{formula})/\sigma_P$. Observed z-
scores are usually greater than 1 and frequently approach 4 (see distribution in Figure 3), indicating that the vapor pressures
estimated from formula-based method is several standard deviations away from the structure-based $\bar{p}_{formula}$.

An interesting (though thought to be coincidental) conclusion from this analysis is that the Daumit and Modified Li methods
are biased from the composite structure-based estimate by roughly equal but opposite amounts. Consequently, an average of
these two methods (Figure 3d) provides a relatively accurate estimate of the vapor pressure of a formula. An ideal formula-
based approach cannot be more accurate that the actual variability in $\bar{p}_{formula}$, so should produce a normal distribution of error:
no bias, with ~68% of estimates within one standard deviation of the formula average, and 95% within two standard deviations.
The combined Daumit-Li method exhibits little to no bias, with 57% of estimates within one standard deviation, 80% within
two standard deviations. This distribution is only a little broader than ideal (i.e. longer tails of high error), so is almost as
precise as a formula-based estimation method can be. Other approaches may be possible to achieve these results (e.g., refitting
coefficients for the Li method), but no such effort is attempted here as they are unlikely to improve the reliability of formula-
based methods, and are no less empirical than combining existing empirical methods.

These results demonstrate that formula-based parameter estimation can provide a representative estimate of vapor pressure for
a given formula, i.e. typical of a large mixture of isomers. However, error in this approach increases if used to estimate the
vapor pressure of a given species. The difference between the formula-based and structure-based estimate of vapor pressure
for a given molecule is frequently several orders of magnitude (Figure 3e), even if using the lowest-error method (the average
of the Daumit and Modified Li methods). This error is significantly higher in the case of toluene oxidation products, further
supporting the conclusion that estimating vapor pressure for these compounds is particularly challenging. Error in estimating
the vapor pressure of an individual molecule using only its formula is approximately the same as $<\Delta p>_{formula}$, the difference in
vapor pressures between isomers (i.e., Figure 1a compared to Figure 3e). This level of error is expected for an optimal formula-
based method, as the lack of structural information as an input means the formula-based method does not distinguish between
isomers so cannot be more precise than differences between them. Considering the average and distribution of error, the
combined Daumit-Li method (modified to consider nitrates) appears to represent a nearly optimal approach to estimating vapor
pressure from a molecular formula.



### 3.3 Isomer differences for Henry's law constant

Like vapor pressure, estimation of HLC can be critical for estimating the partitioning of an atmospheric organic species between vapor and condensed phases. We consequently seek here to address the same issue: whether differences in the HLC

of isomers are larger than the differences between SARS. As noted in the methods, species estimated by either SAR to have HLC $\geq 10^{18}$ M/atm are excluded from analyses.

Similar to $\Delta p$ and $\langle\Delta p\rangle_{formula}$ above, $\Delta HLC$ and $\langle\Delta HLC\rangle_{formula}$ denote here the absolute HLC difference of any isomer pair and the average value of all possible pairs of a given formula, respectively. Isomers are observed to differ in their HLC fairly substantially. Using HWINb, $\langle\Delta HLC\rangle_{formula}$ is less than 3 orders of magnitude, with an overall average of approximately 1.5

log units (Figure 4a). This is slightly lower than the estimate from GROMHE, for which $\langle\Delta HLC\rangle_{formula}$ is less than 4 orders of magnitude, with an overall average of approximately 2 log units. Average variability again obscures the more extreme cases observed across all isomer pairs. The distribution of $\Delta HLC$ for all possible isomer pairs is shown in figure 4b. $\Delta HLC$ sometimes reach up to 4 or 5 log units (Figure 4b). These estimates suggest that $\langle\Delta HLC\rangle_{formula}$ is typically ~1 log unit larger than $\langle\Delta p\rangle_{formula}$, and up to several log units more in extreme cases.

For a given species, HLC estimated with GROMHE and HWINb frequently differ by several orders of magnitude (Figure 4c-d, additional comparisons in Figure S10). We denote the difference between HLC estimation methods for a given species as $\langle\Delta m_H\rangle$. As observed for $\langle\Delta m_p\rangle$, the differences in vapor pressure estimation methods, $\langle\Delta m_H\rangle$ is largest for particle-phase components, especially for the toluene oxidation system. Based on $\langle\Delta m_H\rangle$, it is a reasonable summary of these data that HLC for a given molecule can be estimated to within 2 log units, with a central tendency of ~1 log unit. Overall $\langle\Delta HLC\rangle_{formula}$ is

generally ~1 log unit higher than variability between estimation methods, similar to the case of vapor pressure.





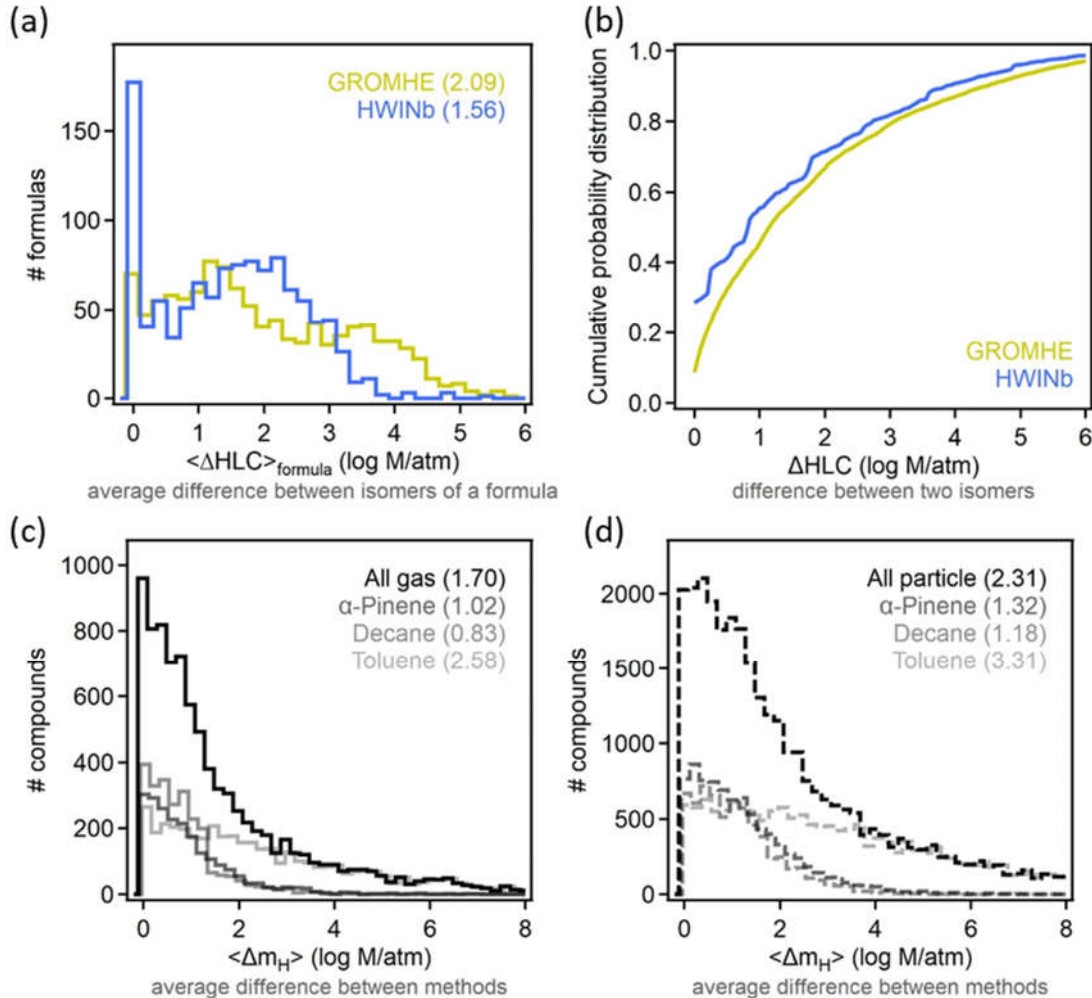

**Figure 4. Differences in Henry's Law Constant (HLC) between isomers and methods. (a) Distribution of <ΔHLC>formula, the average difference between HLC of isomers of a given formula for the two structure-based estimation methods examined. (b) Cumulative probability distribution of ΔHLC, the difference between any two isomers of a given formula for the two methods examined. (c-d) Distribution of absolute differences between structure-based estimates of HLC for a given compound. In the (c) gas and (d) particle phase, which each oxidation system shown in a different lightness. Average values of each distribution are provided in parentheses.**

### 3.4 Estimation of vapor Henry's law constants by formulas

Similar to $\bar{p}_{\text{formula}}$ above, a composite structure-based estimate, $\overline{HLC}_{\text{formula}}$, was computed for each formula as the average value of HLC estimated with both GROMHE and HWINb and for all isomers with that formula. Given the relationship (in log space) observed between volatility (vapor pressure) and solubility (HLC) in previous studies (Hodzic et al., 2014; Lannuque et al., 2018), formula-based estimation of HLC is expected to be achievable. We apply that concept here through a simple



linear regression (Figure 5a) between $\bar{p}_{\text{formula}}$ and $\overline{HLC}_{\text{formula}}$, (i.e., estimated parameter for a formula calculated as the

average for all isomers using all structure-based estimation methods). These parameters are observed to have a linear

relationship ($R^2 = 0.75$) of the form $\log(\overline{HLC}_{\text{formula}}) = -1.15 \log(\bar{p}_{\text{formula}}) - 0.78$, where $\bar{p}_{\text{formula}}$ is in units of *atm* and

$\overline{HLC}_{\text{formula}}$ is in units of *M/atm*. This equation (shown as a purple line in Figure 5b) also effectively describes the relationship

between $\overline{HLC}_{\text{formula}}$ and its vapor pressure estimated using the average of the Modified Li and Daumit methods. Estimation

of HLC in the absence of any structural information (i.e., from formulas alone) is consequently in good agreement with the

average HLC of a formula: exhibiting little bias, within one standard deviation of $\overline{HLC}_{\text{formula}}$ 57% of the time, and two standard

deviations 80% of the time (Figure 5c). This is again approximately as precise as possible for a formula-based method for the

estimation of HLC (as in the case of vapor pressure estimation, there is a longer tail of high error than expected for an ideal

normal distribution). Formula based estimation of HLC therefore appears to provide a reasonable estimate for a typical mixture

of isomers. As in the case of vapor pressure estimation, however, estimating HLC of a single species using its formula is less

reliable, with errors up to 6 orders of magnitude (Figure 5d).

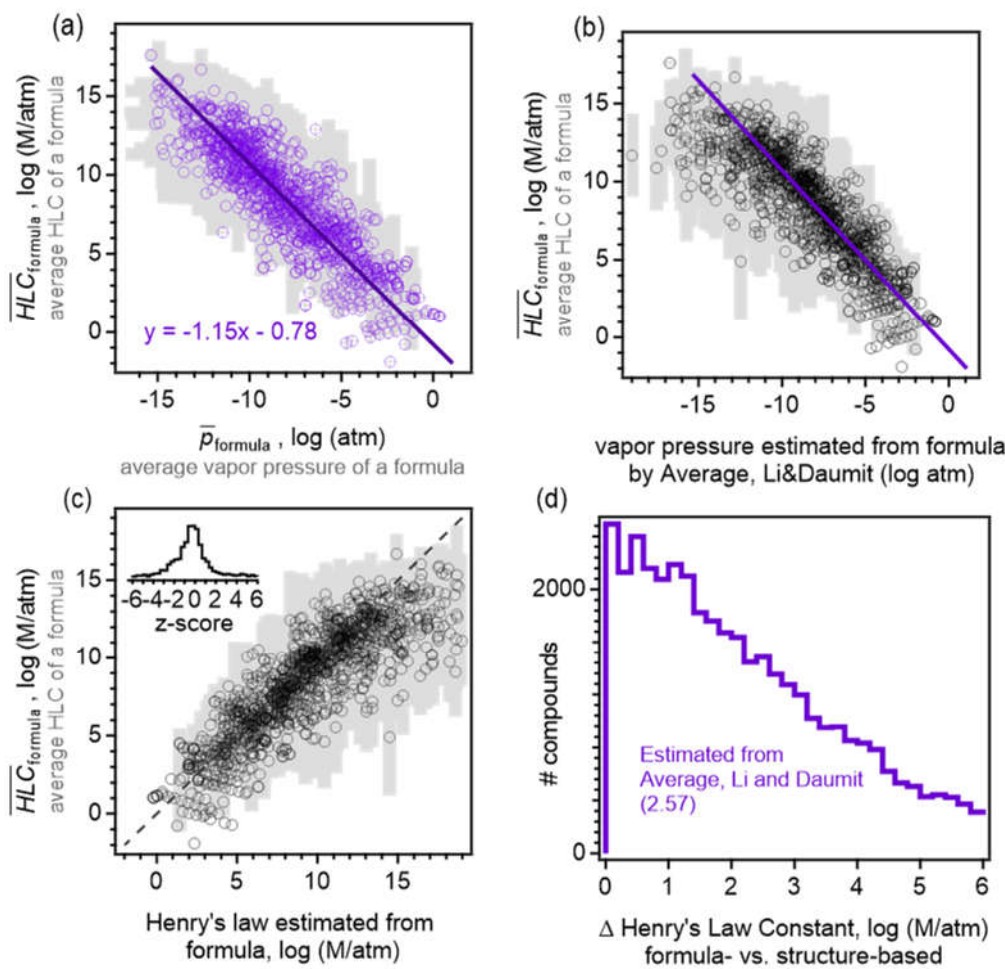

**Figure 5. Comparison between** $\overline{HLC}_{formula}$ **(average HLC of all methods and all isomers) and (a)** $\overline{p}_{formula}$**, the average vapor pressure of a formula with best fit line shown, and (b) the estimated vapor pressure using the formula-based average Daumit-Li method estimate, with same fit line shown in purple. (c) Comparison between** $\overline{HLC}_{formula}$ **and the HLC estimated from vapor pressure calculated from the Daumit-Li method using the best-fit equation shown in panel (a). . Each formula is represented as an open circle at** $\overline{HLC}_{formula}$**, with light-gray bars representing standard deviation of the average,** $\sigma_P$**, to indicate the approximate range. (d) Distribution of error from applying this method to any given compound, with all oxidation system combined, average value provided in parentheses.**

### 3.5 Estimation of $k_{OH}$

The last physicochemical parameter we examine in this work is the rate constant for the reaction between a gas-phase organic compound and the OH radical, $k_{OH}$. The variability in rate constants is also substantially lower for $k_{OH}$ than for other parameters, with nearly all molecules having a rate constant between $10^{-12}$ and $10^{-10}$ cm$^3$ molec$^{-1}$ s$^{-1}$ (as opposed to many orders of magnitude for vapor pressure and HLC). As opposed to the absolute differences in log terms used for the other parameters,



comparisons are consequently more reasonably quantified in terms of relative difference, i.e., $\Delta k = |k_{OH,i} - k_{OH,j}|/k_{OH,i}$, where $i$ in
this work refers to Jenkin and $j$ refers to Kwok and Atkinson. Both methods selected here for structure-based estimation of this
parameter (Jenkin; Kwok and Atkinson) agree that the average difference between isomers, $<\Delta k>_{formula}$, is approximately a
factor of two to three (100-200% relative difference, Figure 6a). In contrast, the two methods tend to differ by only 25-50%
(Figure S11, 75% for toluene products). Differences in the $k_{OH}$ of isomers are therefore significantly larger than apparent
variability in their estimation. Similar to vapor pressure and HLC, for each formula, we compute a composite structure-based
average $\bar{k}_{formula}$ as the average of both methods for all isomers of a given formula. Due to the relatively narrow range of
possible $k_{OH}$, and the significant variability between isomers, $\bar{k}_{formula}$ is not particularly variable across formulas. Most
formulas containing only carbon, hydrogen, and oxygen have rate constants in the range of 2-4 x $10^{-11}$ cm$^3$ molec$^{-1}$ s$^{-1}$, with an
overall average of 2.8 x $10^{-11}$ cm$^3$ molec$^{-1}$ s$^{-1}$ (Figure 6b). Formulas also containing nitrogen (primarily nitrates and
peroxynitrates in this dataset) have an OH reaction rate constant of approximately half this, with a tight distribution centered
around an average of 1.4 x $10^{-11}$ cm$^3$ molec$^{-1}$ s$^{-1}$. These distributions indicate that for any given formula, assuming a constant
$k_{OH}$ within a formula class is almost always accurate to within a factor of 2. It is important to note that, given the dataset used
in this work to calculate these distributions and averages, these results apply only to atmospheric oxidation products and are
not directly applicable to directly-emitted compounds or other atmospheric constituents.

In contrast to the low variability observed for $\bar{k}_{formula}$ the formula-based estimation method developed by Donahue spans a
larger range and typically overestimates $k_{OH}$ (Figure 6c). No correlation is observed between reactivity, $\bar{k}_{formula}$, and vapor
pressure, $\bar{p}_{formula}$ (Figure 6d, $R^2 = 0.15$ within a formula class, $R^2 = 0.02$ in the combined dataset), consistent with results
reported by Lannuque et al. (2018) also showing no clear trend between $k_{OH}$ and $p$. This is in contrast to the Donahue method,
which does predict a strong correlation between these properties (Figure S5, $R^2 = 0.80$). However, Figure 6d does demonstrate
some trends that are in rough agreement with the broad conclusions Donahue et al. (2011) put forth in the manuscript
developing their method: higher volatility compounds react somewhat slower, moderate volatility compounds have rate
constants around 3x$10^{-11}$ cm$^3$ molec$^{-1}$ s$^{-1}$, and lower volatility compounds have slightly higher reaction rates but are likely to
partition to the particle phase and therefore not react quickly with OH.





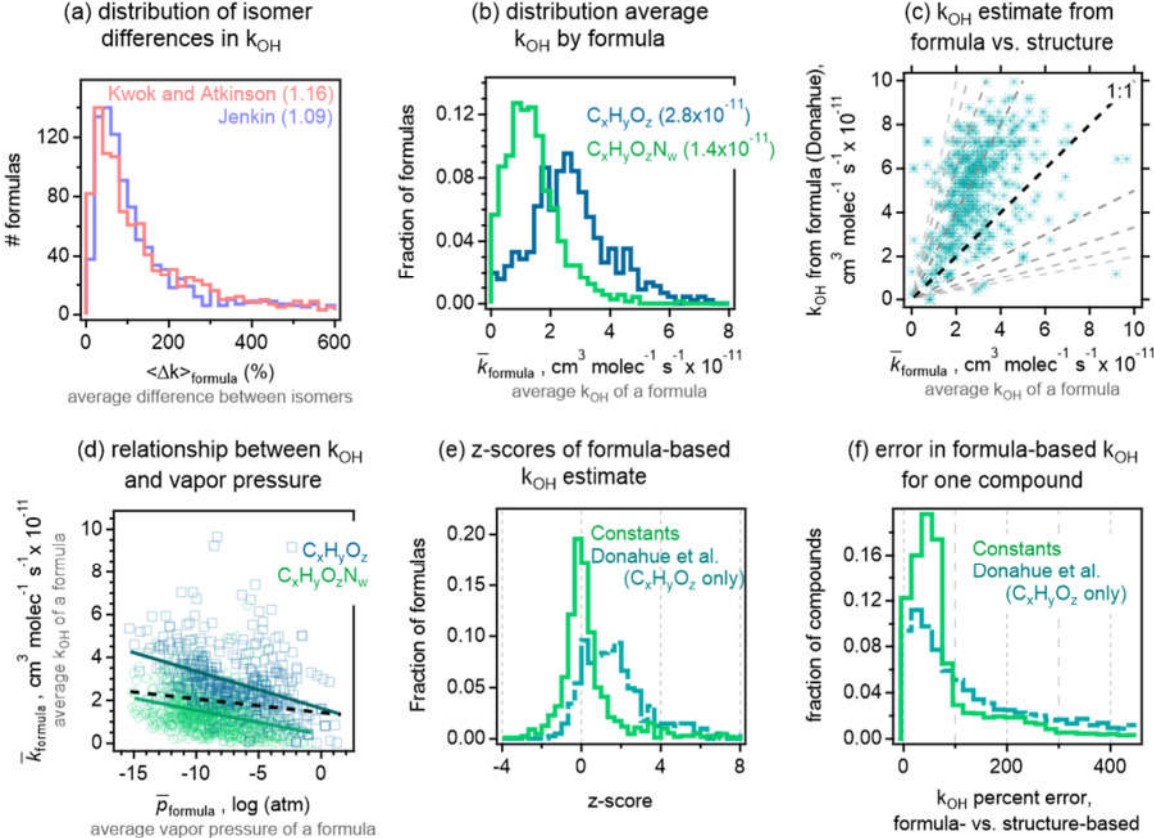

**Figure 6. (a)** Differences in $k_{OH}$ between isomers for the two structure-based estimation methods examined. **(b)** Distribution of $\overline{k}_{formula}$, the average $k_{OH}$ for a given formula calculated as average of both methods for all isomers, shown separately for formulas with and without nitrogen with average value provided in parentheses. **(c)** Comparison of formula-based Donahue estimate to $\overline{k}_{formula}$; dashed lines are 1:1, 1:2, 1:3, etc. **(d)** Comparison of average $k_{OH}$ ($\overline{k}$) to average vapor pressure ($\overline{k}$) for a given formula, separated into formulas with and without nitrogen. Trend lines ($R^2 = 0.15$) shown in the same colors, with trendline for combined set ($R^2 = 0.02$) shown as dashed black line. **(e)** Z-scores of each formula-based methods, calculated as described in Figure 3 and main text. **(f)** Distribution of error from applying this method to any given compound, with all oxidation system combined.

The general overestimation of the Donahue method, coupled with the observation that variability in the $k_{OH}$ of a formula is quite low, suggests an improved method of estimating $k_{OH}$ for a given formula is to simply assume it to have the average value of its formula class (i.e., $2.8 \times 10^{-11}$ and $1.4 \times 10^{-11}$ cm$^3$ molec$^{-1}$ s$^{-1}$ for CHO and CHON, respectively). The distribution of z-scores in Figure 6e indicates that the composite structure-based average $k_{OH}$ for a given compound is usually (71% of the time) within one standard deviation of this average value, and within two standard deviations 86% of the time. This is, once again, approximately as precise as a formula-based method can be. In contrast, the Donahue method frequently overestimates $k_{OH}$ of





a formula by several standard deviations. As in the case of vapor pressure and HLC, formula-based estimation of $k_{OH}$ of an
individual molecule yields errors similar to the average differences between isomers (Figure 6f). However, due to the relatively
low variability of these values, this approach is still typically within a factor of two (100% error) of the average values for each
formula class. These data consequently suggest that approaches to actually estimate the OH reactivity of a gas-phase formula
(including the Donahue approach) are likely to introduce more error than simply a rough assumption of "a few"x10$^{-11}$ cm$^3$
molec$^{-1}$ s$^{-1}$.

**4 Discussion**

In general, structure-based estimation methods tested in this work agree to within approximately half an order of magnitude
for vapor pressure, an order of magnitude for HLC, and <50% for $k_{OH}$. The vapor pressures and Henry's law constants of two
isomers typically tend to differ by a half to a full order of magnitude *more* than the variability in their estimation (i.e.,
differences between SARs), and isomers differ in their $k_{OH}$ by several times the variability in its estimation. Estimation of a
physicochemical parameter from a formula can approximate the average of all relevant isomers within that formula with
reasonable precision and low bias, but application of formula-based methods to an individual molecule from only its formula
introduces higher error. These results support three important conclusions:

(1) Differences in the physicochemical parameters in isomers tend to be larger than differences between estimation
methods, therefore:

(2) When molecular structure is available, its inclusion in the estimation of physicochemical parameters improves the
precision of the estimate, and

(3) Estimating parameters based only on formula is feasible, but is more meaningful if considered as a representative
value for a typical mixture of isomers rather than any species in particular.

We base these conclusions on the methods used in this work to estimate the properties of a formulas in this work: vapor
pressure as the average of EVAPORATION, SIMPOL, and Nannoolal methods; Henry's law constant as the average of
GROMHE and HWINb; and $k_{OH}$ as the average of the Jenkin et al. and Kwok and Atkinson methods. These approaches were
selected based on the accuracy of each SAR as published in previous work, and their publicly available implementations.

One outcome of this work that is of critical importance to the broader atmospheric chemistry community is the demonstration
that different publicly available implementations of a given published structure-based vapor pressure estimation methods (e.g.,
EVAPORATION) may not all produce the same estimates for a given species. While five structure-based methods were
included in this work, three of them have two known publicly available implementations, and in all three cases these two
implementations disagree, often by at least an order of magnitude for large fraction of the species tested. This implies that



while five methods might be nominally used in the literature, there may be up to eight *de facto* methods used (not including manual implementations). Some differences could be due simply to errors in implementing complex parameterizations, but of more fundamental interest is the observation that many differences may be unavoidable outcomes of "extrapolating" SARs. In other words, the complexity of atmospheric species is not always easily described in clear-cut way by the functional groups included in an SAR, and each implementation may parse a structure differently. When possible, estimating a parameter as the average of multiple methods would help minimize the impacts of potential uncertainties in the implementations of each method, in addition to mitigating potential biases or uncertainties of any one method.

Similarly, this work demonstrates the issue that development of empirical techniques such as formula-based estimation methods can be biased by the data used in their development. In particular, the Li et al. (2016) method for estimating vapor pressure from formulas (sometimes known as the "molecular corridors" method) contained few nitrates in its training data, and subsequently exhibits significant bias in the nitrate-heavy systems studied here. We propose a modification to this method to address this limitation, specifically the treatment of each $NO_3$ unit in a formula as an OH unit.

By combining existing methods and new approaches, we also provide in this work new methods for the estimation of vapor pressure, HLC, and $k_{OH}$ for a given molecular formula. The methods below agree with composite structure-based estimates for formula (i.e., average of all structure-based methods for all major isomers) with approximately normally distributed error (with a somewhat longer tail), suggesting they are nearly as precise as possible. The application of the recommended formula-based methods to an individual molecule introduces error comparable to the difference between isomers, which further supports the conclusions that these methods are approximately as precise as such a method can be. Consequently, while estimation of parameters for a formula can be reasonably accomplished, it nevertheless suffers higher uncertainty due to the lack of structural information. It should be noted that the accuracy of formula-based methods is limited by the accuracy of the SARs upon which they are built. This work therefore seeks only to understand the precision, not the accuracy, of formula-based methods in estimating the average SAR-estimated properties of a mixture of isomers of a given formula.

Formula-based estimation methods that are found to estimate average properties of a formula with approximately as high a precision as possible are:

- Vapor pressure – average of the Daumit method and the Li method, after modifying the latter to address its bias for nitrates. Error: roughly 1-2 order of magnitude.
- Henry's law constant – estimated from the above vapor pressure using the linear relationship $\log(HLC) = -1.15 \log(p^0) - 0.78$ (see Figure 5a). Error: roughly 2-3 orders of magnitude.
- $k_{OH}$ – constant depending on whether the formula contains only carbon, hydrogen, and oxygen ($k_{OH} = 2.8 \times 10^{-11}$ cm$^3$ molec$^{-1}$ s$^{-1}$), or also contains nitrogen ($k_{OH} = 1.4 \times 10^{-11}$ cm$^3$ molec$^{-1}$ s$^{-1}$). Error: roughly a factor of 2.



Error is estimated as the ability of the formula-based method to recreate the structure-based estimated property of a formula, not based on the accuracy of the existing SARs on which they are built. Error in vapor pressure and HLC is estimated as a range due in part to its dependence on volatility (more uncertainty at lower volatility) and oxidation system (more error in the aromatic system study). We note that these formula-based estimation methods are empirical, and consequently subject to biases as with other formula-based approaches. We attempt to minimize this issue by developing these methods using the types of atmospherically relevant compounds to which these methods are often applied (oxygenated oxidation products of common precursors), but stress that no empirical method can be fully free of development bias.

To facilitate the adoption of these formula-based approaches, we are including as part of this manuscript the Parameter Estimation for Atmospheric Chemistry (PEACh) package, written in the Igor Pro programming environment (Wavemetrics, Inc.) widely used by the atmospheric chemistry community. PEACh v.1 is included included in the Supplementary Information, and will be updated and maintained as a GitHub repository (github.com/gabrielivw/PEACh). This package implements formula-based estimation by the methods described above. For structure-based estimation, we encourage the practice of averaging multiple SARS for structure-based estimates of properties, and point the reader toward the publicly available implementations used in this work.

## 5 Acknowledgements

This work was supported by the Alfred P. Sloan Foundation Chemistry of the Indoor Environment Program (P-2018-11129). Special thanks to S. Takahama and D. Topping for discussing with us their implementations of structure-based estimation methods, as well as to M. Shiraiwa, J. Kroll, and N. Donahue for discussing with us their groups' formula-based estimation methods.

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
