# Peer review of "Impact of organic molecular structure on the estimation of atmospherically relevant physicochemical parameters"

_Atmospheric Chemistry and Physics, 2020_

## Referee Comment (RC1) · Anonymous Referee #1 · 18 Nov 2020

This is a rigorously and meticulously conducted study about an interesting and important topic. It is also exceptionally well written and beautifully illustrated. I should be accepted after formulating some very general statements more cautiously (see below for details).

Nevertheless, I can't help feeling somewhat disappointed by the study. The reason is that the study (and by inference a part of the atmospheric science community) continues to rely to a very large extent on property prediction methods that are arguably unsuited to the task of predicting phase partitioning of atmospheric oxidation products. The semi-volatile compounds, for which the secondary organic aerosol (SOA) com-

[Figure]

munity requires vapor pressure and Henry's law constant estimates, invariably have multiple functional groups and as such are unquestionably outside of the applicability domain (AD) of the structure activity relationships (SARs) that this community is using. The more functional groups there are on a molecule, the more opportunities there are for these functional groups to interact. Clearly, these intramolecular interactions influence the ability of molecules to interact with other molecules (such as those in SOA or an aqueous aerosol phase). Yet, most of the SARs used in this study were calibrated using empirical data for compounds with very few functional groups, assume additivity of group contributions, and ignore intramolecular interactions. The authors and the community at large in fact is largely aware of these shortcomings, may indirectly even acknowledge them (e.g. lines 50-53, lines 305-306), but then ignores them and keeps using those SARs (out of inertia?). This may have been justifiable as long as there were no alternatives, but that is a stance that is increasingly more difficult to defend in light of the availability of alternative approaches (Wania et al. Novel methods for predicting gas–particle partitioning during the formation of secondary organic aerosol. Atmos. Chem. Phys., 14, 13189–13204, 2014, Wang et al. Uncertain Henry's law constants compromise equilibrium partitioning calculations of atmospheric oxidation products. Atmos. Chem. Phys., 17, 7529–7540, 2017).

The study does not formally test whether the compounds for which predictions are made are within the AD of the prediction methods; the study neither seeks to quantify the prediction error that is incurred when applying a method to a compound outside of its AD (although both should be possible). Instead it attempts to quantify that error by comparing the results of several prediction methods, that are all leveraged well beyond their AD. This approach is unlikely to give a reasonable estimate of the true prediction uncertainty, because these methods are similar in terms of assuming additivity of group contributions and ignoring intramolecular interactions, i.e. the likelihood that they are all similarly biased is considerable. In what really amounts to circular reasoning, three of the vapor pressure estimation methods are assumed to be "the best available" (line 461) by virtue of the similarity of their predictions (line 463). Predictions that are

deemed outliers are simply disregarded in this analysis of uncertainty (e.g. Myrdal and Yalkowsky and EPI, line 454).

I don't suggest that the authors revise their current manuscript by including in the analysis one or the other of the alternative prediction techniques (although they should certainly use those techniques in future endeavors). One recommendation, however, is to make the SMILES codes of the 38,594 atmospherically relevant species representing approximately 1,200 formulae available as a supplementary data file, so that others may be able to conduct a complementary analysis with property prediction techniques, whose ADs actually do comprise those multi-functional compounds.

Line 51, 617: The habit of referring to the Henry's law constant as "solubility" is very unfortunate. The solubility is more appropriately the concentration in a solvent (e.g. water) at saturation. The Henry's law constant (as used in the atmospheric chemistry community) is really the ratio of the solubilities in a solvent and in air (see for example: Cole, J. G., D. Mackay. Correlating environmental partitioning properties of organic compounds: The three-solubility approach. Environ. Toxicol. Chem. 2000, 19, 265-270.) Here on line 50, the reason for the larger divergence in the HLC with increasing number of functional groups occurs because of the lower volatility and not because of a higher water solubility (i.e. the reason for the poorer performance for multi-functional compounds is the same for vapor pressure and HLC).

Incidentally, that is also the reason for the linear relationship between vapor pressure and HLC (Fig. 5a). (However, one has to be very cautious, because the set of compounds investigated here is biased towards chemicals with multiple, polar functional groups, whose activity coefficients in water may not range over a very large range. Large, non-polar chemicals, which are missing from this dataset, would presumably not fit such regressions very well, because of their very high activity coefficients (low water solubilities) in the aqueous phase.)

Line 413: More precisely, this should say: "to understand typical differences in predicted vapor pressure between isomers". As argued above, the applied SARs cannot credibly claim to capture the real differences in the properties of the isomers of multi-functional compounds.

Line 438: General statements like this need to be qualified: "that predicted vapor pressures of the isomers of atmospheric oxidation products differ by "

Line 498: Statement such as this ("the impact of structure is less than variability in estimation methods") are really only valid in the context of the very specific assumptions that the authors made. Would they still be true if the Myrdal and Yalkowsky and EPI predictions had not been excluded from the analysis of variability in estimation methods?

Line 604: Another one of these statement that cannot be made with such generality. Just because two estimation methods, when applied to a particular type of compound (outside of their AD!), agree within two orders of magnitude of each other, you cannot conclude that HLC can be estimated to within 2 log units.

Line 704-707: This is an extremely important qualifying statement. It may be advisable to add: - a clarification that these conclusions are also only based on a very particular type of organic compound, namely atmospheric oxidation products with a fairly limited number of functional group types. (This study even noted substantial differences between the atmospheric oxidation products of different precursor VOCs highlighting that the type of compound plays an important role in this regard.) - a sentence that it is entirely possible that these conclusion could be found to not be valid when a different set of prediction methods (or a different type of compounds) were to be employed, in particular if prediction method were used that do not need to be applied well beyond the AD (somewhat euphemistically termed "extrapolating" SARs on line 715).

Line 732-734, 746-747: Again, I commend the authors for including this qualifying statement.

Line 97: "less exact than structure-based estimation"

Line 232: "EPI" has not been defined as this stage in the manuscript.

Line 435: In scientific writing the plural of "formula" should be "formulae" and not "formulas".

Line 430: "isomer pairs"

Line 516: Rephrase: "The above analysis indicates that vapor pressures are sufficiently different between isomers that . . ."

Line 520: The phrase is not comprehensible: "this approach cannot be more precise that than the impacts of structure on a property"?

Line 567: "more accurate than the actual variability"

Line 614: Delete "vapor"

Line 755: "SARs" not "SARS"

---

## Referee Comment (RC2) · Anonymous Referee #2 · 27 Dec 2020

Isaacman-VanWertz and Aumont conducted comprehensive analysis on important physicochemical properties of organic aerosols (OA), i.e., their vapor pressures, Henry's Law Constants, and gas-phase rate constants, estimated by both structure-based and formula-based methods. They found the predicted property differences between isomers are larger than those caused by different methods. The evaluation of formula-based methods showed reasonable estimations when applied to a mixture of isomers. As molecular structures are often unknown in ambient organic aerosols, formula-based methods are recently developed and adopted to estimate OA proper-ties. This study conducted detailed analysis evaluating formula-based methods and the results, e.g., average of the Daumit method and the modified Li method present-
ing best estimations of vapor pressure, and the development of the PEACh package, provided important information and tools to the aerosol community applying formula-based methods. I am happy to recommend publication of this manuscript in ACP and have only a few minor comments as below:

- While the manuscript is generally written very well, I have several editing comments: (a) define kOH in the abstract. (b) Page 1 Line 29: I suggest modifying "partition between phase and fates". "partition between fates" is not proper. "phase" should be in its plural form. (c) Page 4 Line 97: Change "that" to "than". (d) Page 5 Line 152: Change "condensed phases" to "condensed phase". (e) Page 8 Line 232: Define EPI as Estimation Programs Interface. (f) Page 10 Line 295: What is SB/BK?

- For citations, I suggest adding a few review papers in the Introduction. There was no citation in Lines 22-28 in the Introduction. Ziemann & Atkinson (2012) and Krieger et al. (2012) may be suitable there. Krieger et al. (2018) presenting a data set for validating vapor pressure measurement techniques is suggested to be added around Lines 35-38. Lines 54-72, I understand SIMPOL and EVAPORATION are widely used by the atmospheric community, other estimation methods of the vapor pressure., e.g., Moller et al. (2008), could also be cited and briefly discussed. O'Meara et al. (2014) is also recommended to be added as they also assessed the vapor pressure estimation methods. Page 12 Line 341, you may add Shiraiwa et al. (2014) for "molecular corridors".

- I noticed the authors used fraction of formulas in Fig. 6 as the y-axis. Why in other figures the number of formulas is used instead as the y-axis?

References: Krieger, U. K.; Marcolli, C.; Reid, J. P., Exploring the complexity of aerosol particle properties and processes using single particle techniques. Chemical Society Reviews 2012, 41 (19), 6631-6662.

Krieger, U. K.; Siegrist, F.; Marcolli, C.; Emanuelsson, E. U.; Gøbel, F. M.; Bilde, M.; Marsh, A.; Reid, J. P.; Huisman, A. J.; Riipinen, I.; Hyttinen, N.; Myllys, N.; Kurtén, T.;

Bannan, T.; Percival, C. J.; Topping, D., A reference data set for validating vapor pressure measurement techniques: homologous series of polyethylene glycols. Atmos. Meas. Tech. 2018, 11 (1), 49-63.

Moller, B.; Rarey, J.; Ramjugernath, D., Estimation of the vapour pressure of non-electrolyte organic compounds via group contributions and group interactions. Journal of Molecular Liquids 2008, 143 (1), 52-63. O'Meara, S.; Booth, A. M.; Barley, M. H.; Topping, D.; McFiggans, G., An assessment of vapour pressure estimation methods. Physical Chemistry Chemical Physics 2014, 16 (36), 19453-19469.

Shiraiwa, M.; Berkemeier, T.; Schilling-Fahnestock, K. A.; Seinfeld, J. H.; Pöschl, U., Molecular corridors and kinetic regimes in the multiphase chemical evolution of secondary organic aerosol. Atmos. Chem. Phys. 2014, 14 (16), 8323-8341.

Ziemann, P. J.; Atkinson, R., Kinetics, products, and mechanisms of secondary organic aerosol formation. Chemical Society Reviews 2012, 41 (19), 6582-6605.
* * *

---

## Author Comment (AC1) · 5 Mar 2021

We thank both reviewers for their support and believe our revised manuscript addresses their concerns.

Please see responses below. Reviewer comments are in **green**, our response is in **black**, text excerpted from the revised manuscript is in "quotes", and added language in these excerpts is in *italics*.

**Reviewer Comment 1.**

This is a rigorously and meticulously conducted study about an interesting and important topic. It is also exceptionally well written and beautifully illustrated. I should be accepted after formulating some very general statements more cautiously (see below for details).

We thank the reviewer for their support and recognition of the strengths, limitations, and scope of this manuscript. We hope that our revised manuscript addresses the concerns of the reviewer. We include with the revised manuscript the SMILES and estimated physicochemical properties for all compounds used in this work. We welcome the use of this dataset in continued investigation of this issue, and/or if the reviewer wishes to contact us directly, we would be happy to collaborate on an effort to extend these sorts of analyses to some of the other estimation approaches referenced (e.g., ppLFER).

Nevertheless, I can't help feeling somewhat disappointed by the study. The reason is that the study (and by inference a part of the atmospheric science community) continues to rely to a very large extent on property prediction methods that are arguably unsuited to the task of predicting phase partitioning of atmospheric oxidation products. The semi-volatile compounds, for which the secondary organic aerosol (SOA) community requires vapor pressure and Henry's law constant estimates, invariably have multiple functional groups and as such are unquestionably outside of the applicability domain (AD) of the structure activity relationships (SARs) that this community is using. The more functional groups there are on a molecule, the more opportunities there are for these functional groups to interact. Clearly, these intramolecular interactions influence the ability of molecules to interact with other molecules (such as those in SOA or an aqueous aerosol phase). Yet, most of the SARs used in this study were calibrated using empirical data for compounds with very few functional groups, assume additivity of group contributions, and ignore intramolecular interactions. The authors and the community at large in fact is largely aware of these shortcomings, may indirectly even acknowledge them (e.g. lines 50-53, lines 305-306), but then ignores them and keeps using those SARs (out of inertia?). This may have been justifiable as long as there were no alternatives, but that is a stance that is increasingly more difficult to defend in light of the availability of alternative approaches (Wania et al. Novel methods for predicting gas-particle partitioning during the formation of secondary organic aerosol. Atmos. Chem. Phys., 14, 13189–13204, 2014, Wang et al. Uncertain Henry's law constants compromise equilibrium partitioning calculations of atmospheric oxidation products. Atmos. Chem. Phys., 17, 7529–7540, 2017).

As the reviewer has noted, the need to extrapolate SARs beyond their validated chemical range is an issue we are aware of, and we have revised the manuscript to make this issue more explicitly discussed throughout the manuscript. A few specific examples are excerpted below:

**Added to Section 1 Introduction**

"Various methods exist to estimate volatility (e.g., Barley and McFiggans, 2010; Camredon and Aumont, 2006; Compernolle et al., 2011), HLC (e.g., Meylan and Howard, 1991; Raventos-Duran et al., 2010) and gas-phase reaction rates (e.g., Vereecken et al., 2018). *Though these SARs are frequently used to estimate physicochemical parameters of atmospheric constituents, their application to atmospheric oxidation products often requires extrapolation far beyond the chemical space (i.e., volatility, chemical functionality) used in their development*. Furthermore, many of the molecules present in the atmosphere contain multiple functional groups, and the substituent groups within a complex molecule may not obviously "map" to the groups used to define an SAR *or may interact with neighboring groups in ways not captured by an SAR. This need to extrapolate the volatility and functionality domain of SARs for atmospheric applications leads to higher uncertainty, and previous work has demonstrated that SAR's estimates of vapor pressures, HLC, and gas-phase reaction rates for atmospheric species tend to diverge with increasing number of organic functional groups on the carbon backbone (Raventos-Duran et al., 2010; Valorso et al., 2011)."*

**and**

"An error of half an order of magnitude in vapor pressure for a compound with an estimated saturation concentration near ambient particulate matter concentrations may "move" a compound from mostly in the gas phase to mostly in the particle phase (Compernolle et al., 2011). *Furthermore, uncertainty estimates of half an order of magnitude may be optimistic as recent work has found orders-of-magnitude discrepancies between measured vapor pressures of low-volatility compounds and those estimated by the Nannoolal et al. method (Dang et al., 2019), but data are still limited."*

**Added to Section 3.1 Isomer differences for vapor pressures**

"This phase dependence in the estimated differences in isomer vapor pressures is likely influenced by two complementary issues in applying SARs to this dataset: (a) phase serves as a proxy for volatility, and (b) given that all compounds are products of the same precursors, volatility is decreased primarily by the addition of functional groups and so is a proxy for increased functionality. Consequently, the increased variability in estimated vapor pressures of particle-phase isomers may be due part to the need to extrapolate the SARs toward lower volatility and higher functionality, farther from their well-constrained domains."

**and later:**

"Similar to phase-dependence, system-dependence may be due in part to varying degrees of extrapolating each SAR to functional groups or intramolecular interactions not captured in their development."

**Section 3.3**

"These estimates suggest that  $\langle \Delta HLC \rangle_{formula}$  is typically ~1 log unit larger than  $\langle \Delta p \rangle_{formula}$ , and up to several log units more in extreme cases. This may be due in part to the relatively high uncertainty in

estimating HLC relative to estimating vapor pressure (Hodzic et al., 2014; Wang et al., 2017), as the high uncertainty may contribute to larger variability between estimates for isomers."

One goal of this manuscript (admittedly among several) is to examine the various SARs that are being commonly used in our field. In future work we would be excited to examine some of the other methods described by Wania et al. We have added a discussion of these methods to the introduction, excerpted below. We welcome the rise in more holistic approaches that move beyond some of the limitations of SARs and recognize the cost in developing and operating these tools and the right to charge to re-coup these costs. However, the (perhaps unfortunate) reality is that SARs are likely to see continued use through the near future due to their cost advantage (and, as the reviewer notes, inertia), and so it remains valuable to examine these SARs. Unfortunately, under standard academic licenses, commercial products are substantially more expensive than published SARs, for which there are free publicly available implementations. In the extreme example, SPARC's advertised cost is \$3/calculation (~\$100,000 for the 38,000 SMILES used here); the other commercial products used by Wania and Wang et al. (COSMOtherm, ABSOLV) are cheaper, but costs remain non-negligible. As noted above and below, we encourage any researcher to use the data we provide to examine these next-generation tools, we would be excited to collaborate on that project if the reviewer (or anyone else) is interested, and we would welcome the advice of the reviewer on implementing these approaches in a scalable and affordable way for future work.

**Added to Section 1 Introduction**

To avoid the need to extrapolate SARs and the concomitant uncertainty that arises from this approach, a new generation of tools allows physicochemical properties to be directly estimated using quantumchemistry-based calculations. These tools include commercial products that can directly calculate physicochemical properties (e.g., vapor pressure) or can calculate solvation parameters to estimate partitioning between phases, for instance COSMOtherm (available from Dassault Systèmes, based on COSMO-RS: Klamt, 1995; Klamt and Eckert, 2000) and SPARC Performs Automated Reasoning in Chemistry (available from ARChem LLC, based on: Hilal et al., 2004). In a related approach, a calibrated fit to experimental partitioning data can be developed based on solvation parameters (a poly-parameter linear free energy relationship, or ppLFER), which can in turn be calculated using commercial products like ABSOLV (ACDlabs) (Arp et al., 2008a, 2008b; Wania et al., 2014). By calculating parameters directly from molecular structure, these methods do not suffer the same degree of uncertainty caused by extrapolation beyond the empirically constrained regions of SARs and have been shown to handle multifunctional compounds with no bias and modest increases in uncertainty (Wang et al., 2017). These methods have also been shown to agree well in their estimations of partitioning between vapor and condensed phase organics (related to vapor pressure), but still exhibit large differences in estimations of partitioning of organics into water (related to HLC) (Wang et al., 2017). Quantum-chemistry-based calculations may therefore represent a new approach for estimating partitioning in atmospheric systems (e.g., Wania et al., 2015), but they have not yet seen widespread adoption in the atmospheric science community and so the work presented here focuses on the more commonly used SAR-based approach.

The study does not formally test whether the compounds for which predictions are made are within the AD of the prediction methods; the study neither seeks to quantify the prediction error that is incurred when applying a method to a compound outside of its AD (although both should be possible). Instead it attempts to quantify that error by comparing the results of several prediction methods, that are all leveraged well beyond their AD. This approach is unlikely to give a reasonable estimate of the true prediction uncertainty, because these methods are similar in terms of assuming additivity of group contributions and ignoring intramolecular interactions, i.e. the likelihood that they are all similarly biased is considerable. In what really amounts to circular reasoning, three of the vapor pressure estimation methods are assumed to be "the best available" (line 461) by virtue of the similarity of their predictions (line 463). Predictions that are deemed outliers are simply disregarded in this analysis of uncertainty (e.g. Myrdal and Yalkowsky and EPI, line 454).

As we hope we have made clear, we agree with the reviewers concerns about extrapolation of SARs and are similarly frustrated that there is a dearth of experimental data for lower-volatility atmospheric oxidation products against which to evaluate them. We also recognize that there is a circular aspect to our decision to include Nannoolal, SIMPOL, and EVAPORATION while excluding Myrdal and Yalkowsky (M-Y) and EPI. We do not agree it is entirely circular, however, because it is constrained by the work of Barley and McFiggans (2010) and O'Meara et al. (2014). This work showed Nannoolal with the Nannoolal boiling point method best agreed with experimental data. Consequently, we choose to include SIMPOL and EVAPORATION in part because they agree well with Nannoolal, not solely because all three agree well with each other. Similarly, Barley and McFiggans found high bias in M-Y with Joback and Reid boiling point estimation, which is the M-Y implementation used in the GECKO-A online interface. M-Y with Nannoolal boiling point estimation (available through UManSysProp) may be comparable to the three methods used, but O'Meara found some bias. We have clarified our decisions to include/exclude in the revised manuscript as below. We nevertheless agree that all of these SARs are leveraged well beyond the chemical ranges with which they were developed. The validation work conducted by Barley and McFiggans was limited to higher vapor pressure components, as validation and development of SARs so often are. A better experimental database of some of these parameters for atmospheric oxidation products, and comparison to some of the other methods reference in the future will hopefully allow a better quantification of true error.

**In Section 2.2.4 Myrdal and Yalkowsky:**

"The Myrdal and Yalkowsky SAR has been shown previously to be comparable to, but somewhat less accurate and more biased, than the Nannoolal SAR when the Nannoolal boiling point estimation technique (Nannoolal et al., 2004) is used and substantially biased when Joback and Reid is used (Barley and McFiggans, 2010; O'Meara et al., 2014). The Myrdal and Yalkwosky method is therefore not included in most of the analyses in this work and the GECKO-A and UManSysProp implementations of this SAR are consequently not compared in detailed."

**Added to Section 3.1**

"This assumption is based *in large part* on previous work demonstrating agreement between Nannoolal and experimental data (Barley and McFiggans, 2010; O'Meara et al., 2014), *and the similarity of the other two methods (SIMPOL and EVAPORATION) to Nannoolal*. The EPI and Myrdal and Yalkowsky methods are treated as outliers based on their bias relative to experimental data (shown by Barley and McFiggans, 2010 and O'Meara et al., 2014). By averaging the vapor pressures estimated for each species with Nannoolal, SIMPOL and EVAPORATION methods, we mitigate any biases present in any one method. The average of these three methods provides an average structure-based estimate for a given species, denoted here as  $\bar{p}$ . The methods treated here are of course not exhaustive, but these three methods represent several of the most widely used methods in the field, perform well in comparison to experimental data, and rely on completely independent parameterizations. Other methods that perform well in prior reviews (Barley and McFiggans, 2010; O'Meara et al., 2014), such as the Lee-Kesler method, are not included here either because they are not widely used within the atmospheric field and/or they use the Nannoolal boiling point estimation method (2004) and consequently do not represent a truly independent source of bias or error. "

I don't suggest that the authors revise their current manuscript by including in the analysis one or the other of the alternative prediction techniques (although they should certainly use those techniques in future endeavors). One recommendation, however, is to make the SMILES codes of the 38,594 atmospherically relevant species representing approximately 1,200 formulae available as a supplementary data file, so that others may be able to conduct a complementary analysis with property prediction techniques, whose ADs actually do comprise those multi-functional compounds.

We thank the reviewer for recognizing the scope of the current manuscript. We include with the revised manuscript the full set of SMILES and estimated parameters. We would be glad to see such a complementary analysis happen and, if the reviewer or anyone else is interested in working together on the issue, we would be happy to collaborate on it.

Line 51, 617: The habit of referring to the Henry's law constant as "solubility" is very unfortunate. The solubility is more appropriately the concentration in a solvent (e.g. water) at saturation. The Henry's law constant (as used in the atmospheric chemistry community) is really the ratio of the solubilities in a solvent and in air (see for example: Cole, J. G., D. Mackay. Correlating environmental partitioning properties of organic compounds: The three-solubility approach. Environ. Toxicol. Chem. 2000, 19, 265-270.) Here on line 50, the reason for the larger divergence in the HLC with increasing number of functional groups occurs because of the lower volatility and not because of a higher water solubility (i.e. the reason for the poorer performance for multi-functional compounds is the same for vapor pressure and HLC).

We thank the reviewer for noting this distinction. In the revised manuscript we limit discussion of "solubility" to broad discussions of important parameters and have removed all references to HLC using this term. Incidentally, that is also the reason for the linear relationship between vapor pressure and HLC (Fig. 5a). (However, one has to be very cautious, because the set of compounds investigated here is biased towards chemicals with multiple, polar functional groups, whose activity coefficients in water may not range over a very large range. Large, non-polar chemicals, which are missing from this dataset, would presumably not fit such regressions very well, because of their very high activity coefficients (low water solubilities) in the aqueous phase.)

The reviewer raises a very good point. Prior work by Hodzic et al. on the relationship between these parameters noted that the slope of the line varied by oxidation system, demonstrating that these relationships quantitatively vary by the structures and functionalities of the constituent compounds. As with all the experimentally derived formulations used in this work, extrapolation beyond the compound classes used to develop them can introduce significant error. With that in mind, we agree that the relationship developed here between vapor pressure and HLC should not be extended to non-polar compounds, or perhaps even to non-atmospheric systems without prior evaluation. This point has been added explicitly to the revised manuscript:

**Added to Section 3.4 (discussion of Fig. 5a)**

"Formula-based estimation of HLC therefore appears reasonably precisely capture the estimated HLC of a typical mixture of isomers. However, the average relationship described by this linear fit is necessarily a function of data with which it was generated and previous work exploring this relationship found the slope to vary depending on the oxidized precursor (Hodzic et al., 2014). Consequently, while the relationship shown in Figure 5 represents a reasonable formula-based approach to estimating HLC for a complex mixture of atmospheric oxidation products (moderate-to-low volatility, with multiple functional groups), it should not be extended to other systems (e.g., large, non-polar compounds) without further investigation."

Line 413: More precisely, this should say: "to understand typical differences in predicted vapor pressure between isomers". As argued above, the applied SARs cannot credibly claim to capture the real differences in the properties of the isomers of multifunctional compounds.

We have revised the sentence as suggested (using "estimated" in the place of "predicted" to match the language throughout the manuscript). Clarification that all these values are "estimated" has been added to many locations throughout the revised manuscript.

Line 438: General statements like this need to be qualified: "that predicted vapor pressures of the isomers of atmospheric oxidation products differ by "

We have revised this language here and throughout the manuscript to specify that we are referring to estimated parameters and our dataset is confined to (modeled) atmospheric oxidation products. In addition to several clarifying additions of "estimated" and "oxidation products", we have added in at

least three locations broad clarifying statements about the application of this work beyond atmospheric oxidation products – please see our response to your comments regarding line 704.

Line 498: Statement such as this ("the impact of structure is less than variability in estimation methods") are really only valid in the context of the very specific assumptions that the authors made. Would they still be true if the Myrdal and Yalkowsky and EPI predictions had not been excluded from the analysis of variability in estimation methods?

In the case of the specific statement noted by the reviewer, it is expected to be true even with the inclusion of the other methods, in fact, likely more so. Figure 1 demonstrates that the methods roughly agree on the differences between isomers, while the variability between methods would substantially increase with the inclusion of M-Y and EPI due to their divergence from the other three methods, particularly at the low volatilities referenced in this sentence. Consequently, the impact of structure would be even farther below the variability in estimation methods. However, the broader point of the reviewer is taken, and we agree that  $<\Delta m_x >$  is a function of our specific assumptions and decisions, particularly about which methods to include/exclude. Were all 5 methods included, the volatility region where method-dominated uncertainty crosses over to isomer-dominated uncertainty would be different and might not be the same range of volatility referenced in the next sentence. In addition to small additions of language clarifying that we are discussing a specific subset of methods, we have revised the manuscript to clarify these issues in a few specific locations:

We have added a note regarding issue in the previous paragraphs:

"The  $<\Delta m_p>$  frequency distribution is shown in Figure 2a-b; it is important to note that these distributions are strongly sensitive to the set of methods that are included/excluded."

And have specifically addressed the line referenced by the reviewer.

"At lower vapor pressures however,  $\langle \Delta p \rangle_{formula}$  is not substantially larger than  $\langle \Delta m_p \rangle$ , so the impact of structure is less than variability in estimation methods. Both conclusions are likely insensitive to the specific assumptions about which methods to include in this comparison, as the uncertainty in most estimation methods is generally lowest for high volatility compounds and high for low volatility compounds. However, the transition vapor pressure below which differences between isomers are lost in the uncertainty of our methods is sensitive to the methods included in the comparison. For the three methods included in this comparison, the transition can reasonably be considered to be in the range of  $10^{-10}$  to  $10^{-12}$  atm (c\*  $\approx 10^{-2.5}$  to  $10^{-0.5} \,\mu g/cm^3$ )..."

Line 604: Another one of these statement that cannot be made with such generality. Just because two estimation methods, when applied to a particular type of compound (outside of their AD!), agree within two orders of magnitude of each other, you cannot conclude that HLC can be estimated to within 2 log units.

As discussed above, we have revised this and similar sentences to clarify the assumptions and limitations of the present work.

Line 704-707: This is an extremely important qualifying statement. It may be advisable to add: - a clarification that these conclusions are also only based on a very particular type of organic compound, namely atmospheric oxidation products with a fairly limited number of functional group types. (This study even noted substantial differences between the atmospheric oxidation products of different precursor VOCs highlighting that the type of compound plays an important role in this regard.) - a sentence that it is entirely possible that these conclusion could be found to not be valid when a different set of prediction methods (or a different type of compounds) were to be employed, in particular if prediction method were used that do not need to be applied well beyond the AD (somewhat euphemistically termed "extrapolating" SARs on line 715).

**Added to Section 2 Methods**

"A critical issue to consider throughout this work is that, as with any empirical analysis, the extending results beyond the training data may significantly increase uncertainty. The results herein are most reasonably applied to products of gas-phase atmospheric oxidation, with heavy representation by compounds that are highly oxygenated, are multi-functional, and/or contain nitrate groups."

**Added to Section 3.4 (discussion of Fig. 5a) (see also response to previous comment)**

"Formula-based estimation of HLC therefore appears reasonably precisely capture the estimated HLC of a typical mixture of isomers. *However, the average relationship described by this linear fit is necessarily a function of data with which it was generated and previous work exploring this relationship found the slope to vary depending on the oxidized precursor. Consequently, while the relationship shown in Figure 5 represents a reasonable formula-based approach to estimating HLC for a complex mixture of atmospheric oxidation products (moderate-to-low volatility, with multiple functional groups), it should not be extended to other systems (e.g., large, non-polar compounds) without further investigation.*"

**Added to Section 4 Discussion**

"These conclusions are also necessarily limited to the types of compounds used analyzed in this dataset, namely oxidation products from the gas-phase oxidation of a few representative compound classes. These results can therefore reasonably be extended to oxygenated compounds in complex atmospheric mixtures, particularly with multiple functional groups in which organic nitrogen is the form of nitrates. Extending the conclusions and methods of this work to broader systems will necessarily increase uncertainty."

Line 732-734, 746-747: Again, I commend the authors for including this qualifying statement.

Thank you. As noted, we agree with the reviewer's concerns, and appreciate their understanding that we are working within a knowledge space that suffers substantial unknowns and limitations

Line 97: "less exact than structure-based estimation"

**Corrected**

Line 232: "EPI" has not been defined as this stage in the manuscript.

**Definition added**

Line 435: In scientific writing the plural of "formula" should be "formulae" and not "formulas".

For readability, we have chosen to keep "formulas." However, if the editorial staff would prefer we switch to formulae to preserve journal style, we will accept this decision.

We note a large body of articles within ACP and the related AMT that use "formulas" in the context of chemical/molecular formulas, and in the ACP submission information, which uses both "formulae" and "formulas":

"Mathematical symbols and formulae: in general, mathematical symbols are typeset in italics. The most notable exceptions are function names (e.g. sin, cos), chemical *formulas*, and physical units, which are all typeset in roman (upright) font." (emphasis added)

Examples of papers using "formulas" over the past decade:

Riva, M., Rantala, P., Krechmer, J. E., Peräkylä, O., Zhang, Y., Heikkinen, L., Garmash, O., Yan, C., Kulmala, M., Worsnop, D. and Ehn, M.: Evaluating the performance of five different chemical ionization techniques for detecting gaseous oxygenated organic species, Atmos. Meas. Tech. Discuss., 1–39, doi:10.5194/amt-2018-407, 2018.

Gkatzelis, G. I., Hohaus, T., Tillmann, R., Gensch, I., Müller, M., Eichler, P., Xu, K. M., Schlag, P. H., Schmitt, S., Yu, Z., Wegener, R., Kaminski, M., Holzinger, R., Wisthaler, A. and Kiendler-Scharr, A.: Gas-toparticle partitioning of major biogenic oxidation products: A study on freshly formed and aged biogenic SOA, Atmos. Chem. Phys., 18(17), 12969–12989, doi:10.5194/acp-18-12969-2018, 2018.

Müller, M., Graus, M., Wisthaler, A., Hansel, A., Metzger, A., Dommen, J. and Baltensperger, U.: Analysis of high mass resolution PTR-TOF mass spectra from 1,3,5-trimethylbenzene (TMB) environmental chamber experiments, Atmos. Chem. Phys., 12(2), 829–843, doi:10.5194/acp-12-829-2012, 2012.

Chan, A. W. H., Chan, M. N., Surratt, J. D., Chhabra, P. S., Loza, C. L., Crounse, J. D., Yee, L. D., Flagan, R. C., Wennberg, P. O. and Seinfeld, J. H.: Role of aldehyde chemistry and NOx concentrations in secondary organic aerosol formation, Atmos. Chem. Phys., 10(15), 7169–7188, doi:10.5194/acp-10-7169-2010, 2010.

**Line 430: "isomer pairs"**

**Corrected**

Line 516: Rephrase: "The above analysis indicates that vapor pressures are sufficiently different between isomers that  $\dots$ "

**Corrected**

Line 520: The phrase is not comprehensible: "this approach cannot be more precise that than the impacts of structure on a property"?

**Phrase removed**

Line 567: "more accurate than the actual variability"

Corrected

Line 614: Delete "vapor"

Corrected

Line 755: "SARs" not "SARS"

Corrected

**Reviewer Comment 2**

Isaacman-VanWertz and Aumont conducted comprehensive analysis on important physicochemical properties of organic aerosols (OA), i.e., their vapor pressures, Henry's Law Constants, and gas-phase rate constants, estimated by both structurebased and formula-based methods. They found the predicted property differences between isomers are larger than those caused by different methods. The evaluation of formula-based methods showed reasonable estimations when applied to a mixture of isomers. As molecular structures are often unknown in ambient organic aerosols, formula-based methods are recently developed and adopted to estimate OA properties. This study conducted detailed analysis evaluating formula-based methods and the results, e.g., average of the Daumit method and the modified Li method presenting best estimations of vapor pressure, and the development of the PEACh package, provided important information and tools to the aerosol community applying formulabased methods. I am happy to recommend publication of this manuscript in ACP and have only a few minor comments as below:

We thank the reviewer for their support and hope they find it a useful study for their own future work

- While the manuscript is generally written very well, I have several editing comments:

(a) define kOH in the abstract.

Corrected as "and differ in their rate constant for reaction with OH radicals  $(k_{OH})$ "

(b) Page 1 Line 29: I suggest modifying "partition between phase and fates". "partition between fates" is not proper. "phase" should be in its plural form.

Corrected as "they can partition between phases and may vary in their fates"

(c) Page 4 Line 97: Change "that" to "than".

Corrected

(d) Page 5 Line 152: Change "condensed phases" to "condensed phase".

**Corrected**

(e) Page 8 Line 232: Define EPI as Estimation Programs Interface.

**Corrected**

**(f) Page 10 Line 295: What is SB/BK?**

This designation is used within the referenced work to refer to the Stein and Brown vapor pressure estimation method using boiling point estimated by a modified Baum equation, which is more or less the same method used by EPI. This has been revised as:

"(Barley and McFiggans, 2010, wherein the method *referred to as "SB/BK*" closely approximates the EPI method)."

- For citations, I suggest adding a few review papers in the Introduction. There was no citation in Lines 22-28 in the Introduction. Ziemann & Atkinson (2012) and Krieger et al. (2012) may be suitable there. Krieger et al. (2018) presenting a data set for validating vapor pressure measurement techniques is suggested to be added around Lines 35-38. Lines 54-72, I understand SIMPOL and EVAPORATION are widely used by the atmospheric community, other estimation methods of the vapor pressure., e.g., Moller et al. (2008), could also be cited and briefly discussed. O'Meara et al. (2014) is also recommended to be added as they also assessed the vapor pressure estimation methods. Page 12 Line 341, you may add Shiraiwa et al. (2014) for "molecular corridors".

We thank the reviewer for these suggestions and have included most of these suggestions. With regards to other methods, we have amended the text in a few locations to address this issue. For example:

**Added to Section 3.1**

"This assumption is based in large part on previous work demonstrating agreement between Nannoolal and experimental data (Barley and McFiggans, 2010; O'Meara et al., 2014), and the similarity of the other two methods (SIMPOL and EVAPORATION) to Nannoolal. The EPI and Myrdal and Yalkowsky methods are treated as outliers based on their bias relative to experimental data (shown by Barley and McFiggans, 2010 and O'Meara et al., 2014). By averaging the vapor pressures estimated for each species with Nannoolal, SIMPOL and EVAPORATION methods, we mitigate any biases present in any one method. The average of these three methods provides an average structure-based estimate for a given species, denoted here as  $\bar{p}$ . The methods treated here are of course not exhaustive, but these three methods represent several of the most widely used methods in the field, perform well in comparison to experimental data, and rely on completely independent parameterizations. Other methods that perform well in prior reviews (Barley and McFiggans, 2010; O'Meara et al., 2014), such as the Lee-Kesler method, are not included here either because they are not widely used within the atmospheric field and/or they use the Nannoolal boiling point estimation method (2004) and consequently do not represent a truly independent source of bias or error. "

**- I noticed the authors used fraction of formulas in Fig. 6 as the y-axis. Why in other figures the number of formulas is used instead as the y-axis?**

For the other comparisons using formulas, histograms represent equal numbers of formulas in nearly all cases (i.e., the full set, 1193). This is less true for the Fig. 6b histogram, as there are not the same number of formulas with and without N (more N formulas). Nevertheless, the histogram using number instead of fraction reasonably conveys the same point and has been revised as below.

For the histograms of compounds, this is less true. The set to which Donahue can be applied is subset of the total (i.e., those without N). Furthermore, there are more formulas with N, and these tend to be somewhat more "complex" (more isomers), so the subset to which Donahue can be applied is substantially smaller than the full set. Consequently, a histogram of compounds in absolute terms (attached below) is much harder to draw conclusions from. We have therefore chosen to keep Fig 6f in fraction terms. Similarly, histograms of z-scores are more useful in fractional terms, and are discussed as such in the manuscript, so we have chosen to keep these fractional as well. A discussion to this effect has been added to the caption of this figure in the revised manuscript. We have also clarified in the Methods that roughly two-thirds of formulas contain nitrogen.

Figure 6f in absolute number of compounds